# Coordinated peptidoglycan synthases and hydrolases stabilize the bacterial cell wall

Huan Zhang[1,2], Srutha Venkatesan[1,2], Emily Ng[1] & Beiyan Nan [1] ✉

Peptidoglycan (PG) defines cell shape and protects bacteria against osmotic stress. The growth and integrity of PG require coordinated actions between synthases that insert new PG strands and hydrolases that generate openings to allow the insertion. However, the mechanisms of their coordination remain elusive. Moenomycin that inhibits a family of PG synthases known as Class-A penicillin-binding proteins (aPBPs), collapses rod shape despite aPBPs being non-essential for rod-like morphology in the bacterium *Myxococcus xanthus*. Here, we demonstrate that inhibited PBP1a2, an aPBP, accelerates the degradation of cell poles by DacB, a hydrolytic PG peptidase. Moenomycin promotes the binding between DacB and PG and thus reduces the mobility of DacB through PBP1a2. Conversely, DacB also regulates the distribution and dynamics of aPBPs. Our findings clarify the action of moenomycin and suggest that disrupting the coordination between PG synthases and hydrolases could be more lethal than eliminating individual enzymes.

The peptidoglycan (PG) cell wall, a network of glycan strands and peptide crosslinks, is a hallmark structure of bacteria[1]. PG encases the cytoplasmic membrane, defines cell geometry, and protects the cell against lysis due to its high internal osmotic pressure. As PG is essential for the survival of most bacteria, disrupting PG has become an effective strategy for antibiotic therapies. PG assembly requires multiple synthases, including glycosyltransferases (GTases) that polymerize glycan chains and transpeptidases (TPases) that form peptide crosslinks[2]. During the vegetative growth of most rod-shaped bacteria, two major enzymatic systems carry out the elongation of PG sacculi, the Rod system and class-A penicillin-binding proteins (aPBPs). The Rod system determines rod shape and accounts for most of the PG synthesis during cell growth[3]. The core components of the Rod system include RodA, a SEDS-family GTase, PBP2, a TPase of the class B penicillin-binding proteins (bPBPs), and MreB, a bacterial actin homolog that orchestrates the actions of RodA and PBP2[4–7]. In contrast, aPBPs possess both GTase and TPase activities and their functions do not depend on the Rod system[4,8–10]. While aPBPs assemble PG in different environments, repair PG defects, and regulate cell diameter[3,11,12], their absence rarely abolishes cell survival or rod-like morphology[3,13]. However, the inhibitors of aPBPs usually trigger rapid collapse of rod shapes[14,15].

The insertion of new PG strands requires local hydrolysis of the existing PG network[2,16–18]. PG hydrolases, including lytic transglycosylases, amidases, and endo/carboxypeptidases, are difficult to study in most model organisms. On the one hand, these enzymes are highly redundant in most bacteria, where strains lacking single, or several hydrolases usually do not show significant growth or morphological defects. On the other hand, as uncontrolled hydrolytic activities could compromise PG integrity and cause cell lysis, it is difficult to observe highly activated PG hydrolysis during normal growth[16,19]. To maintain the integrity of the cell wall, PG hydrolases must coordinate with synthases in time and space[16,18–20]. However, even less is known about the mechanisms of their functional coordination.

*Myxococcus xanthus* is a rod-shaped, Gram-negative bacterium that possesses both the Rod system and aPBPs. In response to chemical signals, such as glycerol, vegetative cells of *M. xanthus* thoroughly degrade their PG and transform into spherical spores within a few hours[21–23]. When spores germinate, emerging cells rebuild rod-shaped PG without preexisting templates[23]. Such morphological transitions make *M. xanthus* an ideal organism to study PG degradation and assembly[24]. The Rod system is essential for the de novo establishment of rod shape during *M. xanthus* spore germination[23,25]. In contrast, spherical spores can still germinate into rods in the presence of

[1]Department of Biology, Texas A&M University, College Station, TX 77843, USA. [2]These authors contributed equally: Huan Zhang, Srutha Venkatesan. ✉e-mail: bnan@tamu.edu

cefsulodin and cefmetazole, the inhibitors of aPBPs. However, these newly established rods retrogress to spheres before cell division[23]. This observation suggests that aPBPs are important for the maintenance of rod-like morphology albeit the mechanism remains unknown.

In this work, we identified DacB, a PBP4-family D-Ala-D-Ala endo/carboxypeptidase, as a major PG hydrolase that collapses the rod shape of *M. xanthus* by degrading PG, especially at cell poles. We found that whereas either the inhibition or absence of aPBPs is sufficient to enrich DacB to cell poles, only the inhibition, but not absence, of aPBPs triggers the loss of rod shape in vegetative cells. We demonstrate that inhibited PBP1a2, an aPBP, promotes the binding between DacB and PG, and thus reduces the diffusion of DacB. Conversely, DacB also regulates the dynamics and distribution of aPBPs. Our results elucidate the mutual regulation between PG synthases and hydrolases, which plays central roles in the maintenance of cell integrity.

## Results

### The inhibition of aPBPs triggers rapid collapse of rod shape

To study how aPBPs and the Rod system support rod shape, we treated vegetative *M. xanthus* cells using moenomycin (8 μg/ml) and mecillinam (100 μg/ml). Whereas moenomycin is a phosphoglycolipid that specifically inhibits the GTase activity of aPBPs but does not affect RodA, the GTase of the Rod system[10], mecillinam is a ß-lactam that specifically inhibits the Rod system by blocking the Tpase activity of PBP2[26,27]. After 2 h, 72.7% ($n = 967$ cells) of moenomycin-treated cells became spherical. Agents that inhibit PG synthesis or disrupt PG were suspected to induce sporulation of *M. xanthus*[28]. To distinguish whether the spherical cells that resulted from moenomycin treatment are spores or vegetative cells that had lost rod shape, we treated these cells with sonication. Compared to the glycerol-induced spores that 91.7% ($n = 2515$ cells) are resistant to sonication[23], the moenomycin-induced spheres are collapsed vegetative cells because only 5.3% ($n = 1034$ cells) of them remained after sonication. Although prolonged moenomycin treatment turned 100% cells ($n = 1356$ cells) into spheres, we did not observe mass cell lysis. Instead, cell concentration continued to increase during the treatment (Fig. 1a, b). Different from L-forms that usually generate new cells through irregular budding[29], the moenomycin-induced spheres produced daughter cells of the same size (Fig. 1a), indicating that these cells proliferate through binary fission. Currently, we are investigating how *M. xanthus* cells divide as spheres. In contrast to moenomycin, 2-h of mecillinam treatment only caused minor bulging near the centers of cells and none of the treated cells ($n = 988$ cells) lost rod shape (Fig. 1a). After prolonged incubation,

these bulges became more pronounced while the cells still maintained rod-shaped near their poles (Fig. 1a). It is possible that aPBPs could substitute PBP2 in the presence of mecillinam, especially near cell poles. Nevertheless, our data indicate that the inhibition of aPBPs by moenomycin is sufficient to trigger rapid collapse of rod shape in *M. xanthus*.

### Cells maintain rod shape in the absence of all aPBPs

As moenomycin inhibits aPBPs specifically, we investigated whether the cells that lack certain aPBPs still maintain rod shape. *M. xanthus* encodes three aPBPs, including two homologs of PBP1a (PBP1a1, ORF K1515_11805[30] and PBP1a2, ORF K1515_08245) and one putative PBP1c (ORF K1515_25460). The strains that lack one, two, or all three aPBPs were all viable and only the absence of PBP1a1 caused moderate slow-down of growth in early exponential phase (Fig. 2a). All seven mutants maintained rod shape. Nevertheless, compared to wild-type, mutant cells were moderately, but significantly shorter and wider (Fig. 2b, c). Thus, while being dispensable for both growth and rod-like morphology, all aPBPs regulate cell dimensions. Compared to wild-type cells, the strains lacking aPBPs, especially PBP1a1 and PBP1a2, showed significantly enhanced resistance against moenomycin, which is evidenced by higher numbers of cells that remained rods after a 2-h treatment by 8 μg/ml moenomycin (Fig. 2d). However, even in the absence of all three aPBPs, 15.3% ($n = 1217$ cells) of cells still lost rod shape (Fig. 2d). This result suggests minor and nonspecific toxicity of moenomycin on *M. xanthus*. Nevertheless, our data indicate that it is the inhibition, but not absence of aPBPs that triggers the collapse of rod shape.

### Moenomycin promotes PG hydrolysis by DacB

Blocked aPBP activities by moenomycin could cause the loss of rod shape. However, this hypothesis cannot explain the fact that cells completely lacking aPBPs are still rods. Alternatively, the inhibition of aPBPs could activate or alter the distribution of certain PG hydrolases, which results in mis-regulated PG degradation. In a previous attempt to label PG using a fluorescent D-amino acid, TAMRA 3-amino-D-alanine (TADA)[23,31], we found that deleting *dacB*, a gene encoding a PBP4-family D-Ala-D-Ala endo/carboxypeptidase, improved the incorporation of TADA significantly[23] (Fig. 3a, b). Thus, consistent with its peptidase activity, DacB actively removes TADA from the peptide stem of PG.

To visualize PG degradation by DacB, we labeled the entire PG layer of cells by inducing the germination of *M. xanthus* spores in the presence of TADA. Both the newly generated wild-type and ΔdacB cells showed homogeneous incorporation of TADA (Fig. 3a). After 1 h of moenomycin treatment, the labeled wild-type cells lost fluorescence near evenly throughout the PG layer, whereas bright loci of TADA remained at the poles in ΔdacB cells (Fig. 3c). Taken together, moenomycin activates the hydrolytic activity of DacB, especially at cell poles.

Eliminating the peptidase activity of DacB, by either deleting the *dacB* gene or replacing its conserved catalytic serine residue[32] with alanine (DacB^S75A, Fig. S1a), resulted in moderate, but statistically significant, increase in cell width and decrease in cell length (Fig. 4a, b), similar to the phenotypes caused by the absence of aPBPs (Fig. 2b–d). These observations suggest that DacB contributes to the maintenance of rod shapes. To test if DacB participates in the degradation of rod shape in moenomycin-treated cells, we grew cells in liquid medium to OD_600 1.0 and tested their response to moenomycin (8 μg/ml). After a 2-h incubation, compared to the wild-type cells only 27.8% remained rods, 61.3% of the ΔdacB cells ($n = 1008$ cells) and 57.5 ($n = 1095$ cells) *dacB^S75A* cells retained rod shapes, indicating significantly higher resistance (Fig. 4c). We then expressed photo-activatable-mCherry (PAmCherry)-labeled DacB ectopically using a vanillate-inducible promoter[33] in addition to the endogenous *dacB* gene. Induced by

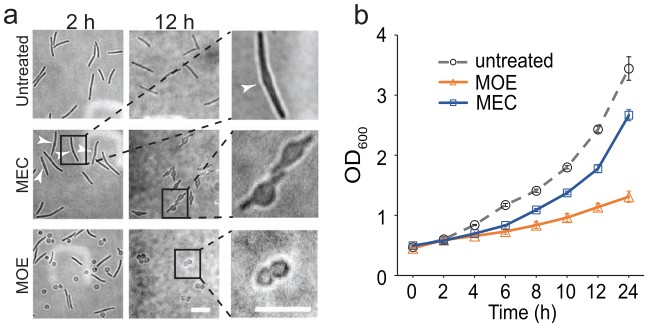

**Fig. 1 | The inhibition of aPBPs abolishes rod shape of *M. xanthus*.**
**a** Moenomycin (MOE, 8 μg/ml) that inhibits aPBPs induces rapid loss of rod shape of *M. xanthus*, whereas mecillinam (MEC, 100 μg/ml), an inhibitor of the Rod system, does not. White arrows point to the bulges induced by MEC. Scale bars, 5 μm.
**b** Because the cell concentrations continue to increase during treatment, neither MOE nor MEC induce mass cell lysis. Cells were inoculated at OD_600 0.5 and growth measured for 24 h from six biological replicates. Data are presented as mean values ± SD. Source data are provided as a Source Data file.

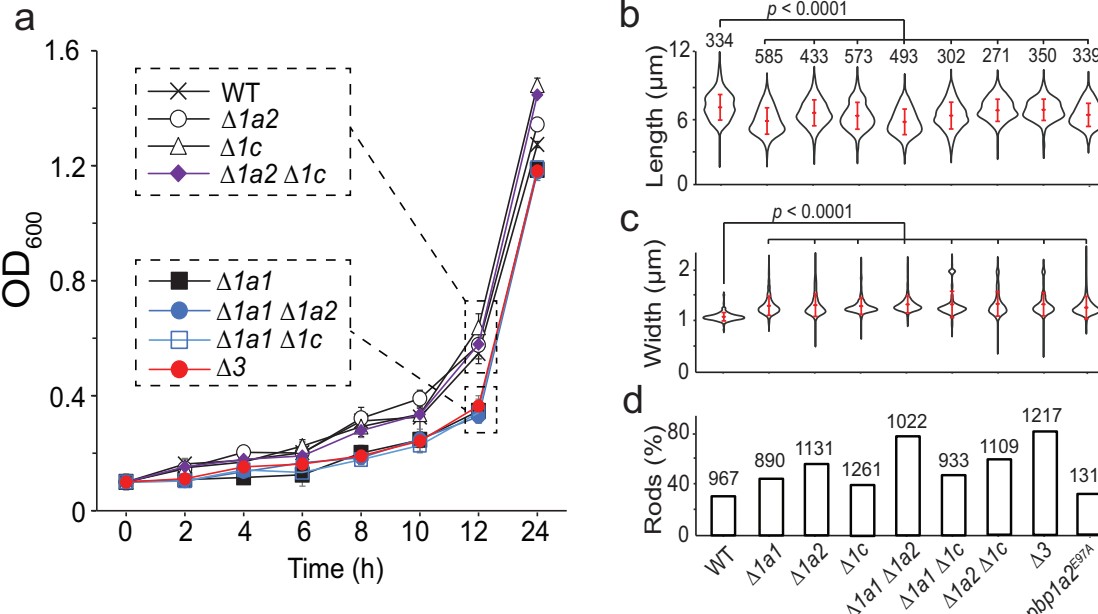

**Fig. 2 | Cells maintain rod shape in the absence of all aPBPs. a** Cells lacking one, two, or all three (*Δ3*) aPBPs are all viable and only the absence of PBP1a1 causes moderate slow-down of growth. Cells were inoculated at $OD_{600}$ 0.1 and growth measured for 24 h from six biological replicates. Data are presented as mean values ± SD. **b–d** While dispensable for the establishment of rod shape, all three aPBPs regulate the length (**b**) and width (**c**) of cells. Compared to the wild-type cells, the absence of aPBPs but not the expression of an enzymatic inactive PBP1a2 increased the numbers of rod-shaped cells after MOE treatment (8 μg/ml, 2 h) (**d**). *p* values were calculated using a one-way ANOVA test between two unweighted, independent samples. Whiskers indicate the 25th–75th percentiles and red dots the median. The total number of cells analyzed is shown on top of each plot. Source data are provided as a Source Data file.

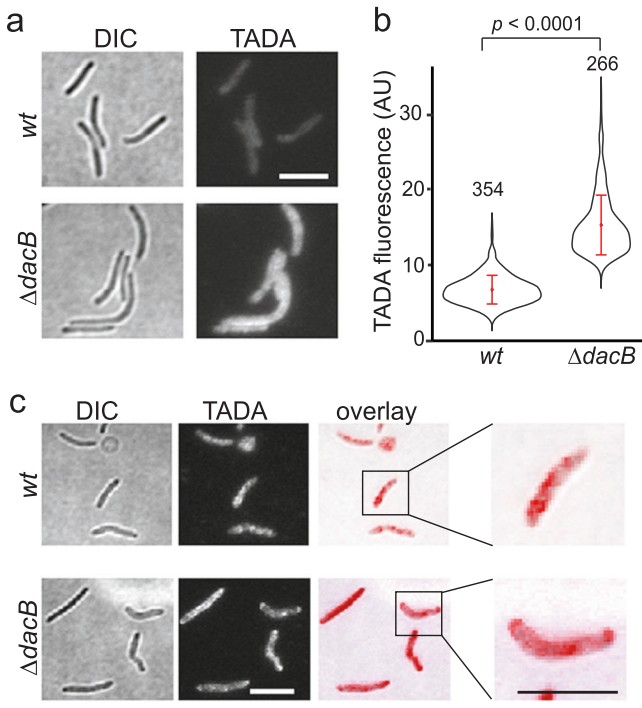

**Fig. 3 | Moenomycin promotes PG hydrolysis by DacB, especially at cell poles. a** The absence of DacB enhances the efficiency of TADA incorporation. **b** Average fluorescence intensities of TADA in vegetative wild-type and *ΔdacB* cells. *p* values were calculated using a one-way ANOVA test between two unweighted, independent samples. Whiskers indicate the 25th–75th percentiles and red dots the median. The total number of cells analyzed is shown on top of each plot. Source data are provided as a Source Data file. **c** Moenomycin induces near-even loss of TADA signal in the wild-type cells while the *ΔdacB* cells retain TADA at their poles. Scale bars, 5 μm.

200 μM vanillate, the cells overexpressing DacB were hyper-sensitive to moenomycin (14.1% rods, *n* = 1046 cells), whereas the ones harboring the empty vector showed sensitivity similar to the wild-type (25.2% rods, *n* = 1172 cells) (Fig. 4c, S1). Interestingly, the overexpression of DacB did not alter cell dimensions significantly (Fig. 4a–c).

### Moenomycin alters the spatial distribution of DacB

To investigate the effects of moenomycin on DacB, we fused a DNA sequence encoding PAmCherry to the endogenous *dacB* gene in wild-type *M. xanthus*. The resulted strain expressed DacB-PAmCherry as a full-length fusion protein that did not show detectable degradation (Fig. S1). The introduction of PAmCherry did not significantly affect cell morphology or the strain's sensitivity to moenomycin, indicating that the tagged DacB protein is functional (Fig. 4). We used a 405-nm excitation laser to activate the fluorescence of a few labeled DacB particles randomly in each cell and quantified their localization using a 561-nm laser. Along the long axis of cells, we loosely defined a region within three pixels (480 nm) from each end of cell as a "pole" and the rest of the cell as the "nonpolar region". By this definition, poles account for ~20% of the cell surface in a typical *M. xanthus* cell (~5 μm long, 1 μm wide). We found that in untreated cells, 28.0% (*n* = 1955 particles) of DacB particles were detected at cell poles, suggesting that DacB distributes near evenly in the membrane. Consistent with the degradation pattern of PG (Fig. 3c), DacB aggregated at poles (47.6%, *n* = 938 particles) in the presence of moenomycin (Fig. 5a, S2).

### Moenomycin enhances PG-binding of DacB

To understand how moenomycin promotes PG degradation by DacB, it is essential to monitor its local peptidase activity in vivo, which is technically difficult. To circumvent this hurdle, we attempted to use the binding between DacB and PG to proximate instantaneous DacB activity in live cells. Compared to individual synthases and hydrolases, the existing PG meshwork is a large and relatively stationary substrate. In *Escherichia coli*, individual particles of aPBPs distribute between

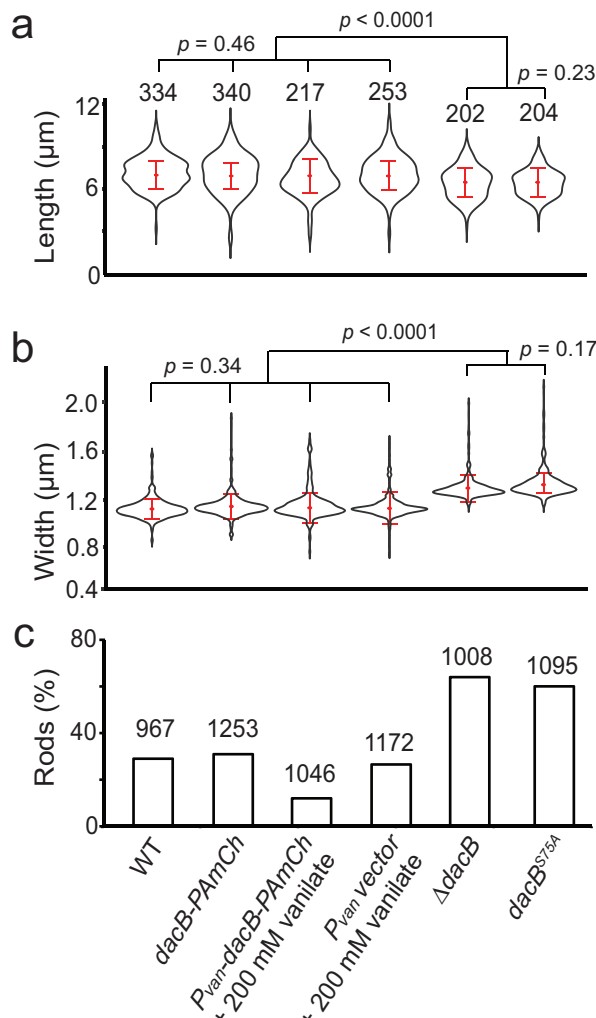

**Fig. 4 | DacB regulates cell morphology and sensitivity against moenomycin.** **a**, **b** DacB moderately regulates cell morphology. *p* values were calculated using a one-way ANOVA test between unweighted, independent samples. Whiskers indicate the 25th–75th percentiles and red dots the median. **c** DacB significantly regulates cell sensitivity against moenomycin. The deletion and overexpression of *dacB* increases and decreases the numbers of rod-shaped cells, respectively, after moenomycin treatment (8 μg/ml, 2 h). The PAmCherry label and vanillate, the inducer for *dacB* overexpression, does not affect moenomycin sensitivity. The total number of cells analyzed is shown on top of each plot. Source data are provided as a Source Data file.

immobile and mobile populations and their inhibitors increase their mobility[9]. Similarly, we expect that the mobility of DacB will decrease when it binds to PG. We imaged single DacB-PAmCherry particles at 10 Hz using single-particle tracking photo-activated localization microscopy (sptPALM) under the highly inclined and laminated optical sheet (HILO) illumination[25,34,35]. Using this setting, only a thin section of each cell surface that was close to the coverslip was illuminated. To analyze the data, we only chose the fluorescent particles that remained in focus for 4–12 frames (0.4–1.2 s). As free PAmCherry particles diffuse extremely fast in the cytoplasm, entering and exiting the focal plane frequently, they usually appear as blurry objects that cannot be followed at 10-Hz close to the membrane[25]. For this reason, the noise from any potential degradation of DacB-PAmCherry was negligible. Using a 405-nm laser (0.3 kW/cm², 0.1 s) to activate PAmCherry and a 561-nm laser for imaging, we detected 1476 DacB-PAmCherry particles from 217 cells. In contrast, we only detected 27 particles from 369 cells

without activation and 25 particles from 449 unlabeled cells using the same setting. Thus, the noises from the background, and autoblinking, the spontaneous switch-on of PAmCherry, were also negligible. DacB particles displayed two dynamic patterns, immobile and diffusion. The immobile particles remained within a single pixel (160 nm × 160 nm) before photo-bleach and the mobile ones displayed typical diffusive behavior (Fig. S2). In wild-type, untreated vegetative cells, 13.1% ($n = 1955$ particles) of DacB particles were immobile and the diffusion coefficient of the mobile population is $2.72 \times 10^{-2} \pm 3.7 \times 10^{-3}$ μm²/s ($n = 1698$ particles) (Fig. 5b, c).

Sporulation provides a scenario where PG hydrolysis activities are upregulated. During glycerol-induced sporulation when cells thoroughly degrade PG, the expression of the *dacB* gene is not regulated[36]. To test if DacB degrades PG during sporulation, we used the length/width ratio (L/W) of cells to quantify the sporulation progress and found that the *ΔdacB* cells did not degrade PG efficiently at poles and thus formed spores slowly. By contrast, the overexpression of DacB-PAmCherry significantly accelerated the rod-to-sphere transition (Fig. S3). Thus, compared to vegetative growth, DacB is activated during glycerol-induced sporulation, especially at cell poles. Consistently, DacB concentrated at poles (59.4%, $n = 2290$ particles) in the presence of glycerol, its immobile population increased to 25.6% ($n = 2290$ particles) and the diffusion coefficient of the mobile population decreased to $2.2 \times 10^{-2} \pm 4.0 \times 10^{-3}$ μm²/s ($n = 1704$ particles) (Fig. 5a–c).

To further confirm the connection between DacB dynamics and its binding to PG, we labeled DacB$^{S75A}$ with PAmCherry and expressed it as the sole source of DacB using the native *dacB* locus and promoter. DacB$^{S75A}$-PAmCherry accumulated as a full-length protein which did not show detectable degradation (Fig. S1). We expected that the mutation on its active site would prevent DacB$^{S75A}$-PAmCherry from binding to PG and thus increase its mobility. Surprisingly, DacB$^{S75A}$-PAmCherry particles displayed reduced mobility, where 16.0% ($n = 2046$ particles) of particles were immobile and the diffusion coefficient of the mobile population was $2.1 \times 10^{-2} \pm 4.9 \times 10^{-3}$ μm²/s ($n = 1719$ particles). These data strongly suggested that despite to the loss of peptidase activity, DacB$^{S75A}$ could bind PG with enhanced affinity. To test this hypothesis, we expressed the periplasmic domains (amino acids 21 – 494) of DacB and DacB$^{S75A}$ in *E. coli* and purified the recombinant proteins. After a 1-h incubation with purified *M. xanthus* PG and a brief centrifugation, both proteins precipitated with PG. Compared to the wild-type protein, a significantly bigger population of DacB$^{S75A}$ enriched in the pellet, indicating increased binding to PG (Fig. 5d). The mechanism for this enhanced binding remains to be elucidated. Nevertheless, as multiple residues contribute to PG-binding in DacB[32], other residues could overcompensate the substitution of the active serine. Taken together, increased immobile population and decreased diffusion coefficients of DacB particles strongly correlates with their enhanced binding to PG.

Compared to untreated cells, during moenomycin-induced PG degradation, the immobile population of DacB increased to 22.6% ($n = 938$ particles) and the diffusion coefficient of the mobile population decreased to $1.9 \times 10^{-2} \pm 4.9 \times 10^{-3}$ μm²/s ($n = 722$ particles) (Fig. 5b, c). Thus, moenomycin enhances PG-binding of DacB. In contrast, mecillinam that does not trigger immediate PG degradation, did not show significant effects on either the dynamics or distribution of DacB (Fig. 5a–c).

To determine whether moenomycin promotes PG-binding by DacB throughout the whole cell or specifically at cell poles, we studied the dynamics of DacB in nonpolar regions. Like its effects on whole cells, moenomycin increased the immobile population to 27.2% ($n = 484$ particles) in nonpolar regions and decreased the diffusion coefficient of the mobile population to $1.7 \times 10^{-2} \pm 4.0 \times 10^{-3}$ μm²/s ($n = 352$ particles). Thus, moenomycin promotes PG-binding by DacB throughout the whole cell (Fig. 5e, f).

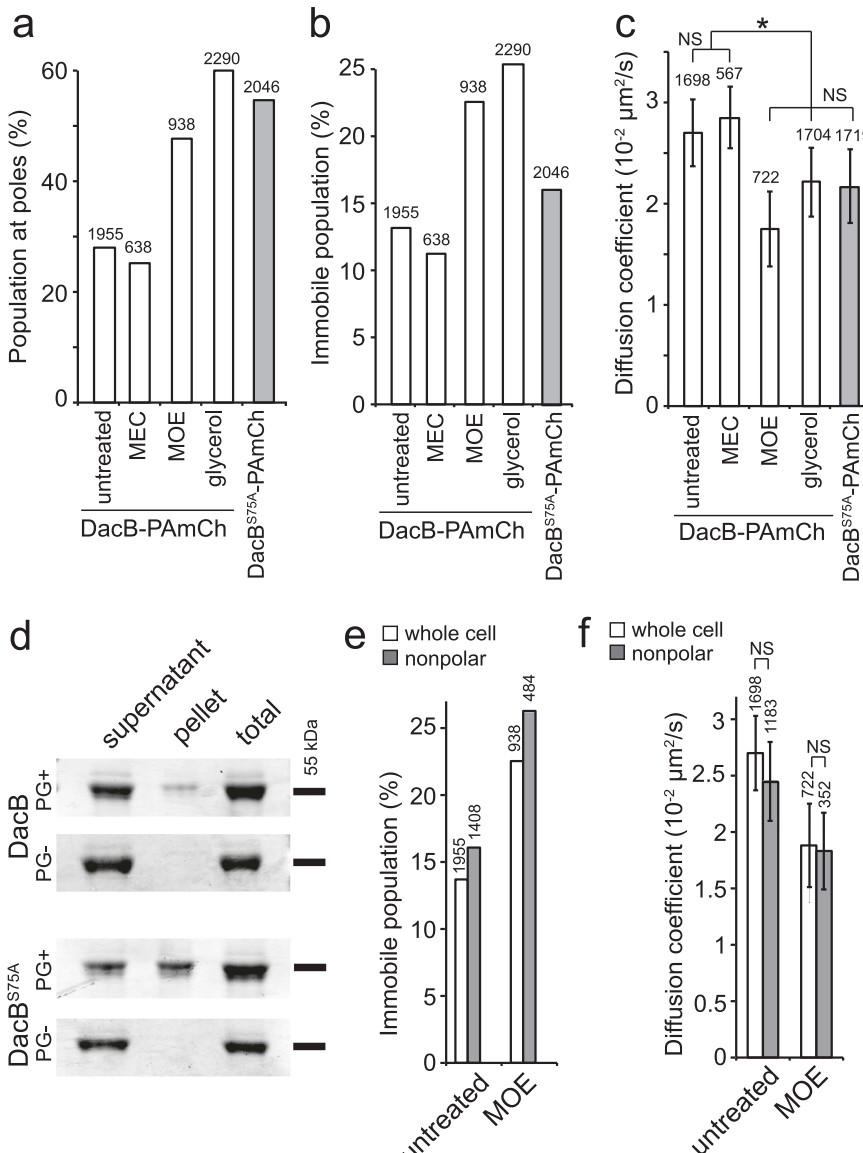

**Fig. 5 | The binding to PG decreases DacB mobility.** Moenomycin (MOE) and glycerol-induced sporulation trigger PG degradation by DacB, especially at cell poles. **a** Both MOE and glycerol enrich DacB particles to cell poles. **b, c** MOE and glycerol reduce the mobility of DacB particles. In contrast, mecillinam (MEC) does not affect the distribution or dynamics of DacB (**a–c**). **d** Whereas the purified periplasmic domains of DacB and DacB[S75A] both co-precipitate with PG, DacB[S75A] binds to PG with higher affinity. This experiment was repeated independently three times with similar results. Uncropped gel images are provided in the Source Data file. Thus, compared to DacB-PAmCherry, DacB[S75A]-PAmCherry particles display lower mobility (**b, c**). **e, f** Moenomycin increases the population of immobile DacB (**e**) and reduces the diffusion coefficient of mobile DacB (**f**) throughout the whole cell. The number of particles analyzed is shown on top of each bar. Mean and standard deviation were calculated through 1000 bootstrap samples of single-particle tracks. * Significant. NS non-significant. Mean values with a difference >0.005 are considered significantly different. Source data are provided as a Source Data file.

## Moenomycin activates DacB through PBP1a2

We hypothesized that moenomycin could activate DacB through aPBPs. To test this hypothesis, we first investigated the cellular distribution of DacB-PAmCherry particles in the strains that lack aPBPs. Like the effect of moenomycin, the absence of one aPBP is sufficient to enrich DacB to cell poles (Fig. 6a). However, none of these mutant lost rod shape (Fig. 2). Thus, although all three aPBPs regulate the distribution of DacB, the polar enrichment of DacB is not sufficient to dismantle rod shape.

We then investigated if aPBPs regulate PG-binding by DacB. In untreated vegetative cells, the absence of each aPBP did not alter the mobility of DacB (Fig. 6b, c, S4). Thus, uninhibited aPBPs do not regulate PG-binding by DacB. In response to moenomycin, DacB mobility reduced in *Δpbp1a1* and *Δpbp1c* cells. Strikingly, moenomycin failed to

reduce DacB mobility in the absence of PBP1a2 (Fig. 6b, c). Taken together, moenomycin-inhibited PBP1a2 promotes PG-binding by DacB.

Is the inactivation of PBP1a2 sufficient to activate PG hydrolysis by DacB? To answer this question, we replaced the conserved catalytic glutamate residue[26] of PBP1a2 with alanine using site-directed mutagenesis and expressed the mutant protein PBP1a2[E97A] as the sole source of PBP1a2 using the native *pbp1a2* locus and promoter. The alanine substitution did not affect the expression or stability of PBP1a2[E97A] (Fig. S5). Cells expressing PBP1a2[E97A] and the ones lacking PBP1a2 displayed indistinguishable morphological defects, including decreased cell length and increased width. However, the loss of active site did not enhance cell resistance against moenomycin, suggesting that the mutant PBP1a2 still binds to moenomycin (Fig. 2).

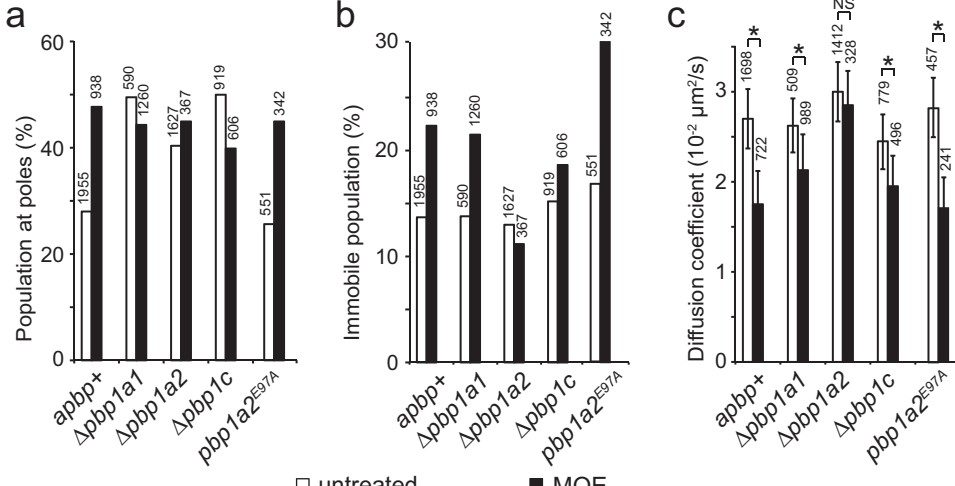

Fig. 6 | Moenomycin activates PG hydrolysis by DacB through PBP1a2. a All three aPBPs and their activities are required for the even distribution of DacB. Their absence and inhibition by MOE both enrich DacB to cell poles. b, c MOE does not reduce DacB mobility in the absence of PBP1a2 but does so in the presence of PBP1a2^E97A. The number of particles analyzed is shown on top of each bar. Mean and standard deviation were calculated through 1000 bootstrap samples of single-particle tracks. * Significant. NS non-significant. Mean values with a difference >0.005 are considered significantly different. Source data are provided as a Source Data file.

Unlike the *Δpbp1a2* cells, the untreated cells expressing PBP1a2^E97A did not enrich DacB at cell poles (Fig. 6a) or reduce DacB mobility (Fig. 6b, c), indicating that the binding between moenomycin and PBP1a2 is required to promote PG-binding by DacB (see discussion). In contrast, moenomycin reduced DacB mobility through PBP1a2^E97A (Fig. 6b, c).

### DacB regulates the distribution and dynamics of aPBPs
The coordination between PBP1a2 and DacB suggests that DacB could also regulate the activity and distribution of aPBPs. To test this hypothesis, we expressed PAmCherry-labeled aPBPs using their endogenous loci and promoters. All three fusions accumulated as stable proteins in *M. xanthus* and unlike the aPBP deletion mutants, cells expressing these fusion proteins maintained wild-type cell morphology and sensitivity to moenomycin (Fig. S6a, b), indicating that PAmCherry-labeled aPBPs are functional. Deletion of the *dacB* gene affected the distribution of all three aPBPs (Fig. 7a). Like DacB, aPBPs displayed two dynamic patterns, immobile and diffusion (Fig. 7b, S6c). We compared the effects of *dacB* deletion and moenomycin on these aPBPs. As the absence of DacB and the presence of moenomycin showed similar effects on the distribution and diffusion of PBP1a1 and PBP1c, DacB could activate both aPBPs. (Fig. 7). Because the deletion of *dacB* promoted PBP1a2 diffusion, (Fig. 7b, c), DacB could also activate PBP1a2 by facilitating its binding to PG. Taken together, our data suggest that DacB could activate all three aPBPs.

### Discussion
It has long been speculated that PG hydrolases associate with synthases in many functions that maintain the integrity of PG[16,19,20]. Although most of the PG synthases and hydrolases have been identified, their functional coordination is less understood. In this report, we dissected the functional coordination between PBP1a2, a bifunctional PG synthase, and DacB, a PG hydrolase. We found that when PBP1a2 binds moenomycin, it promotes PG-binding by DacB and thus activates PG hydrolysis. Why have bacteria developed such a seemingly self-destructive mechanism, and why moenomycin-inhibited PBP1a2 activates DacB but the catalytic inactive PBP1a2^E97A does not? We speculate that moenomycin mimics a growing PG strand and thus locks PBP1a2 into a catalytic active conformation[26]. In contrast, PBP1a2^E97A is in the catalytic inactive conformation. Under

physiological conditions, catalytic active PBP1a2 could recruit DacB to the PG assembly sites, where DacB breaks existing peptide crosslinks and thus facilitates the insertion of newly synthesized murein strands. The fact that moenomycin-bound PBP1a2 recruits DacB to PG, rather than forming a diffusive PBP1a2-DacB complex, suggests that no matter whether these two proteins interact directly, their coordination is mediated by PG.

A major hurdle in studying the coordination between aPBPs and PG hydrolases is that both categories of enzymes are redundant, and many single mutations do not cause clear phenotypes. Furthermore, PG synthases and hydrolases may not form stable complexes, especially in the absence of PG, that are readily detected by biochemical methods. As inhibited PBP1a2 activates the hydrolytic activities of DacB and cells overexpressing DacB are more sensitive to moenomycin than the wild-type ones, our results support the hypothesis that PBP1a2 and DacB may only interact transiently in *M. xanthus* and that one PBP1a2 may activate multiple DacB molecules. In addition, we found that cells lacking aPBPs and DacB are viable, whereas the changes in their coordination cause cell lysis. For these reasons, genetic and biochemical approaches alone might not be sufficient for studying the complex coordination among these enzymes. This research has provided a method to study such coordination before the identification of the coupling mechanisms. If the localization and dynamics of one protein respond to the inhibitor or absence of another, it is sufficient to conclude that the functions of these proteins coordinate.

Recent research on rod-like morphology has emphasized the roles of the cylindrical section where the Rod system plays major roles in the assembly of PG[2,17,37]. In contrast, the contribution of cell poles, where PG remains metabolically inert after division[38], is less appreciated. As moenomycin induces PG hydrolysis at cell poles, we propose that poles may lose inertness under certain stresses. Particularly, our findings provide an explanation for the lytic effects of aPBP inhibitors. As the Rod system is generally excluded from cell poles[39], when aPBPs are inhibited, poles become void of PG synthases. The lack of PG repair in combination with the activation of hydrolysis, causes rapid PG degradation at poles, which results in sudden loss of rod shape. In contrast, the cylindrical surfaces of cells are maintained by both the Rod system and aPBPs. Thus, when one system is inhibited, the other is still able to repair PG damages[3].

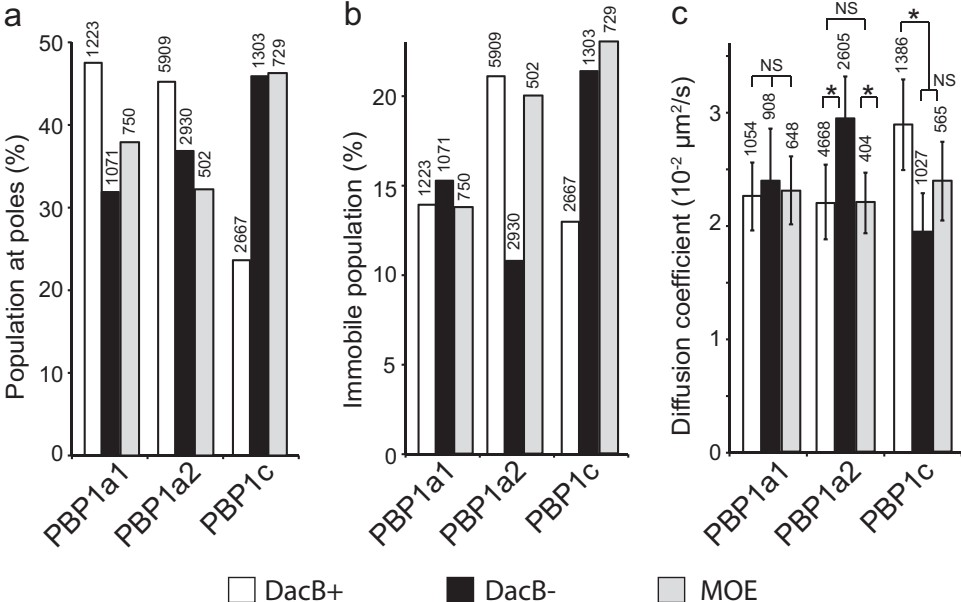

**Fig. 7 | DacB and moenomycin regulate the dynamics and distribution of aPBPs.** The effects of the absence of DacB and the presence of moenomycin on the distribution (**a**) and dynamics (**b**, **c**) of individual aPBPs. The number of particles analyzed is shown on top of each bar. Mean and standard deviation were calculated through 1000 bootstrap samples of single-particle tracks. * Significant. NS, non-significant. Mean values with a difference greater than 0.005 are considered significantly different. Source data are provided as a Source Data file.

Despite exhaustive data on how antibiotics exert action on their targets, we know very little about how these drugs actually kill bacteria[40]. For instance, ß-lactams specifically inhibit the TPase activity of PBPs. However, rather than the blockage of PG assembly, the actual lethal effect of ß-lactams is inducing the degradation of un-crosslinked glycan strands, which are produced by the uninhibited GTases[41]. Similarly, blocking the GTase activities of aPBPs is not likely the direct bactericidal effect of moenomycin. Like *M. xanthus*, although wild-type *Bacillus subtilis* is sensitive to moenomycin, a strain lacking all four aPBPs is still viable and produces PG with only minor defects[4,13]. In this study, we provide evidence that moenomycin inhibits PBP1a2, which in turn, regulates the activity and distribution of DacB and eventually causes cell lysis. We propose that the bactericidal action of moenomycin is inducing an imbalance between aPBPs and PG hydrolases. Our results suggest that disrupting the coordination between PG synthases and hydrolases could be an effective strategy for antibacterial therapies.

## Methods

### Strains and growth conditions

Vegetative *M. xanthus* cells were grown in liquid CYE medium (10 mM MOPS pH 7.6, 1% (w/v) Bacto™ casitone (BD Biosciences), 0.5% yeast extract, and 8 mM $MgSO_4$) at 32 °C, in 125-ml flasks with vigorous shaking, or on CYE plates that contains 1.5% agar. All genetic modifications on *M. xanthus* were made on chromosome. Deletion and insertion mutants were constructed by electroporating *M. xanthus* cells with 4 μg of plasmid DNA. Transformed cells were plated on CYE plates supplemented with 100 μg/ml sodium kanamycin sulfate or 10 μg/ml tetracycline hydrochloride. DacB, PBPs, and their variants were labeled with PAmCherry at their C-termini by fusing their genes to a DNA sequence that encodes PAmCherry through a AAGGAGTCCGGCTCCGTGTC CTCCGAGCAGCTGGCCCAGTTCCGCTCCCTGGAC (KESGSVSSEQLAQ FRSLD) linker. All constructs were confirmed by PCR and DNA sequencing. The expression and stability of PAmCherry-labeled proteins were determined by immunoblotting using an anti-mCherry antibody (Rockland Immunochemicals, Inc., Lot 46705) and a Goat anti-Rabbit IgG (H+L) Secondary Antibody, HRP (Thermo Fisher Scientific, catalog # 31460). The blots were developed with Pierce™ ECL Western Blotting Substrate (Thermo Fisher Scientific REF 32109) and MINI-MED 90 processor (AFP Manufacturing). *M. xanthus* strains used in this study are listed in Table S1. In all imaging experiments using fluorescence microscopy, moenomycin and mecillinam were added at 4 μg/ml and 100 μg/ml, respectively.

### Sporulation and TADA-labeling

To induce sporulation, glycerol was added to 1 M when liquid cell culture reaches $OD_{600}$ 0.1–0.2. After rigorous shaking overnight at 32 °C, the remaining vegetative cells were eliminated by sonication, and spores were purified by centrifugation (1 min, 15,000 × *g* and 4 °C). The pellet was washed three times with water. To prepare TADA-labeled vegetative cells, purified spores were diluted to $OD_{600}$ 0.5 into 1 ml of liquid germination CYE (CYE medium supplemented with 150 μM TADA, additional 0.2% casitone and 1 mM $CaCl_2$) and incubated in an 18-mm test tube at 32 °C, with vigorous shaking.

### Protein expression and purification

DNA sequences encoding amino acids 21–494 of DacB and DacB[S75A] were amplified by polymerase chain reaction (PCR) and inserted into the pET28a vector (Novogen) between the restriction sites of *Eco*RI and *Hind*III. The resulted plasmids were transformed into *E. coli* strain BL21(DE3). Transformed cells were cultured in 20 ml LB (Luria-Bertani) broth at 37 °C overnight and used to inoculate 1 L medium which contains 1.0% (w/v) Bacto Tryptone, 0.5% Bacto yeast extract (BD Biosciences), 0.5% NaCl and 1.0% glucose. Protein expression was induced by 0.1 mM IPTG (isopropyl-h-d-thiogalactopyranoside) when the culture reached an $OD_{600}$ of 0.8. Cultivation was continued at 16 °C for 10 h before the cells were harvested by centrifugation at 6000 × *g* for 20 min. Proteins were purified using a NGC™ Chromatography System (BIO-RAD) and a 5-ml HisTrap™ column (Cytiva)[42,43]. Purified proteins were concentrated using Amicon™ Ultra centrifugal filter units (Millipore Sigma) with a molecular cutoff of 10 kDa and stored at −80 °C.

### PG purification and DacB binding assay

*M. xanthus* cells were grown until mid-stationary phase and harvested by centrifugation (20 min, 6000 × *g*). Supernatant was discarded and

the pellet was resuspended and boiled in 1x PBS with 5% SDS for 2 h. SDS was removed by repetitive wash with MilliQ water and centrifugation (21,000 × g, 10 min, 20 °C). Purified PG from 100 ml culture was suspended into 1 ml 1× PBS and stored at −20 °C[44]. For the binding assay, 10 µl of purified PG was mixed with 10 µl of purified protein (4 mg/ml) and incubated at 25 °C for 1 h. 5 µl of the mixture was taken out and used as "total protein". The remaining 15-ul mixture was subjected to centrifugation (21,000 × g, 10 min, 25 °C), and the supernatant was removed. The pallet was resuspended into 15 µl 1x PBS. The total protein, pellet, and supernatant samples were mixed with equal volumes of 2X loading buffer and 5 µl of each mixture was applied to SDS PAGE. Protein bands were stained using Coomassie brilliant blue.

### Microscopy analysis

For all imaging experiments, 5 µl cells were spotted on an agarose pad. For the treatments with inhibitors, inhibitors were added into both the cell suspension and agarose pads. The length and width of cells were determined from differential interference contrast (DIC) images using a MATLAB (MathWorks) script[25,45]. DIC and fluorescence images of cells were captured using a Hamamatsu ImagEM X2™ EM-CCD camera C9100-23B (effective pixel size 160 nm) on an inverted Nikon Eclipse-Ti™ microscope with a 100x 1.49 NA TIRF objective. For sptPALM, *M. xanthus* cells were grown in CYE to $4 \times 10^8$ cfu/ml and PAmCherry was activated using a 405-nm laser (0.3 kW/cm²), excited and imaged using a 561-nm laser (0.2 kW/cm²). Images were acquired at 10 Hz. For each sptPALM experiment, single PAmCherry particles were localized in at least 100 individual cells from three biological replicates. sptPALM data were analyzed using a MATLAB (MathWorks) script[25,45]. Briefly, cells were identified using DIC images. Single PAmCherry particles inside cells were fit by a symmetric 2D Gaussian function, whose center was assumed to be the particle's position[25]. Particles in consecutive frames were considered to belong to the same trajectory when they were within a user-defined distance of 320 nm (two pixels). Particles that explored areas smaller than 160 nm × 160 nm (within one pixel) in 0.4-1.2 s were considered as immobile. For the mobile particles, mean square displacements (MSDs) were calculated. Time-averaged MSD (TAMSD) of each mobile particle was calculated according to the standard method and fit to $\log (\text{TAMSD}) = \log (4D) + \alpha \cdot \log (\Delta t)$, where $D$ is the diffusion coefficient and $\Delta t$ is time lapse[46]. Based on a previous simulation, diffusive particles on *M. xanthus* cell surface feature $\alpha$ values between 0 and 1.5[25]. In untreated cells, most PAmCherry-labeled particles belonged to this category (93.4% for DacB-PAmCherry, $n = 1995$ particles; 89.7% for DacB$^{S75A}$-PAmCherry, $n = 2046$ particles; 92.7% for PBP1a1-PAmCherry, $n = 1223$ particles; 94.6% for PBP1a2-PAmCherry, $n = 5909$ particles; and 95.1% for PBP1c-PAmCherry, $n = 2667$ particles). For simplicity, we considered all the mobile particles of these proteins diffused and determined their diffusion coefficient ($D$) from a linear fit to the first four points of the MSD using a simpler formula $\text{MSD} = 4D\Delta t$[9,25,45]. Mean and standard deviation were calculated through 1000 bootstrap samples of single-particle tracks. Mean values with a difference >0.005 are considered significantly different.

### Reporting summary

Further information on research design is available in the Nature Portfolio Reporting Summary linked to this article.

## Data availability

All data supporting the findings of this study are available within the paper and its supplementary information. Due to their large sizes, raw microscopy images and videos are available from the corresponding author upon request. Processed cell dimension and single-particle tracking data are provided in the Source Data file, which is provided with this paper. Source data are provided with this paper.

## Code availability

The codes used for cell dimension measurement and single-particle dynamic analysis are described in ref. 23 and deposited in the GitHub repository[45].

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

## Acknowledgements

We thank Xue Fan for the advice on data analysis, Drs. Michael Van Nieuwenhze and Yen-Pang Hsu for providing TADA, and Joshua Pettibon for critical reading of this manuscript. This work is supported by the National Institute of Health R01GM129000 to B.N.

## Author contributions

H.Z., S.V., and B.N. designed the study. B.N. wrote the manuscript. H.Z. and S.V. carried out imaging and performed data analysis. B.N. performed site-directed mutagenesis, protein expression and purification, and PG-binding assay. H.Z., S.V., E.N., and B.N. contributed to strain construction. All authors read and approved the manuscript.

## Competing interests

The authors declare no competing interests.
