## [Peer Review File · Nature Communications]

Coordinated peptidoglycan synthases and hydrolases stabilize
bacteria cell wallReviewer #1 (Remarks to the Author):

In this manuscript, the authors explore mechanisms involved in moenomycin resistance in *Myxococcus xanthus*. From other bacteria, it is known that moenomycin inhibit the glycosyltransferase activity of aPBPs involved in peptidoglycan (PG) synthesis thereby causing cell lysis. Here, the authors find that moenomycin sensitivity depends on DacB, a D-Ala-D-Ala carboxypeptidase. Moreover, it is reported that inhibition of PBP1a2 by moenomycin may activate DacB thereby causing excessive PG hydrolysis and cell lysis. By contrast, lack of PBP1a2 is reported to not activate DacB. This is a potentially highly interesting manuscript, which may help to elucidate how moenomycin causes cell lysis. However, as outlined in details below, the manuscript suffers from the authors not including the original data in their figures. Also, no statistical test to evaluate for significant differences are included. Also, simple control experiments and follow up experiments are missing.

Major comments:

1. line 58-61: Description of aPBPs in *M. xanthus*: From the description of the aPBPs it is not clear whether they are essential. In other words, please clarify whether the rod-shaped cells turned spherical are viable. Also include details on how the aPBPs were inhibited.
2. line 78: Do all the moenomycin treated cells eventually become spherical and lyse? Please include these data to clarify.
2. line 84: How are spores formed in untreated cells? Please clarify.
3. Line 94 and figure 1B, figure 2C: What is meant by "delay in growth"? Please show growth curves for all the mutants, so that the growth defect can be clearly seen. Please also include tests for significant difference and number of replicates analyzed.
4. Line 95 and figure 1C: Please include data on cell length and cell width for all the relevant mutants to document the statement that the mutants maintain rod-shape. Please also include tests for significant differences and number of replicates analyzed.
5. line 110-116: Why are 2µg/ml of moenomycin used in this experiment and 4µg/ml in the experiments in figure 1. This makes a comparison difficult. Also for these experiments: Please show growth curves in figure 2C. Moreover, to strengthen the idea that lack of DacB gives resistance to moenomycin, please include data to show the MIC of the WT, the dacB mutant and the pbp1a2 mutant (see below), and the strain overexpressing dacB. Also, please include quantitative data on cell morphology of these different strains. These data are essential to evaluate the connection between pbp1a2, dacB and moenomycin.
6. line 117-123 and figure 2D: Please include before and after images of the moenomycin treated cells in 2D. The standard deviations in the diagrams in 2D, right are large. Are any of the observed differences significant? Please also include tests for significant difference and number of replicates analyzed.
7. line 124-130 and figure 2c: Please include a western to show that DacB-mCherry accumulates as a full-length fusion protein. These data are essential to be able to evaluate all the localization analyzes.
8. line 124-130 and figure 2c and figure 3: If the full-length fusion protein accumulates, then please include data from simple regular low-resolution epifluorescence microscopy to show the overall localization of DacB-mCherry in the different strains analyzed and +/- moenomycin. These data are essential to be able to evaluate whether the single molecule tracking reflects the overall DacB-mCherry distribution/localization.
9. All the single molecule tracking experiments: Please include a vector control, i.e. mCherry without DacB, to document that the particles tracked are the fusion protein and not random background. Also, to be able to evaluate the tracking data, please include trajectories of e.g. 10 particles per subset. Also, try to superimpose all the trajectories for a given experiment on a schematic cell. These data are essential to be able to evaluate the various statements. Many data are reported on mobile/immobile fractions and diffusion coefficients. For all these numbers, there are no tests included for significant differences and number of experiments. Please also include tests for significant differences and number of replicates analyzed. Also, please make clear which concentration of moenomycin is used in the different experiments.

10. line 171-174: It is assumed that hydrolytic activity of DacB-mCherry is reflected in the mobility of single particles. No evidence is included to directly support this conclusion. Because the interpretation of many of the presented data rely on this assumption, it is essential to include data to support this assumption. This could for instance done by including the localization of a DacB active site mutant.

11. line 180-184: The prediction from the data presented is that a pbp1a2 mutant should be resistant to moenomycin. Is that the case? Also, an active site mutant of PBP1a2 should be resistant to moenomycin and have no effect on DacB. Is that the case? These additional data are important to evaluate the various inferences made by the authors.

12. line 192- 201 and figure 4: Include westerns to document the accumulation of the various fusion proteins. These data are essential to be able to evaluate all the localization analyzes. If the full-length fusion proteins accumulate, then please include data from simple regular low-resolution epifluorescence microscopy to show the overall localization of the PBPa-mCherry proteins in the different strains analyzed and +/- moenomycin. These data are essential to be able to evaluate whether the single molecule tracking reflects the overall distribution/localization of these proteins.

Minor comments:

1. line 36: what is meant by "for the majority of PG growth"? Maybe replace with "synthesis during cell growth". Also, replace "which" with "and".

2. line 39: What does "function outside of..." mean? Function independently?

3. line 56: Replace gram with Gram (it is a name).

Reviewer #2 (Remarks to the Author):

**The coordination between penicillin-binding protein 1 1a (PBP1a) and the hydrolytic peptidase DacB determines the integrity of bacterial cell poles
Zhang et al. have studied the connection between the activity of a class A Penicillin-binding protein, Pbp1a, and a cell wall hydrolase, DacB, in Myxococcus xanthus. The propose a model in which PBP1a influences the spatial distribution of DacB within the cell wall, such that inhibition of Pbp1a leads to stronger DacB activity at cell poles, which is detrimental for cells and leads to cell shape changes and ultimately lysis. A regulation of hydrolytic activity via a class A PBP would be a novel mechanism.**

However, the study has some serious flaws. Most importantly, it is claimed that the antibiotic "moenomycin" specifically inactivated the transglycosylase of Pbp1a, but not of RodA. This is based on a single publication where a crystal structure of Pbp2 from E. coli bound to moenomycin was reported (Loveringh et al., Science 2007). Because Class A PBPs show quite some sequence divergence, it is not clear if specificity for PBP2 from a different bacterium can be taken face value for specificity for Pbp1a. Additionally, moenomycin is a mimic for a short well wall filament with some modifications, so there is no real reason why it should not bind to RodA. Given that the deletion of the gene encoding for Pbp1a does not result in a change of rod shape, but the depletion of RodA does, and so does addition of moenomycin, it is doubtful whether the chemical really only inhibits Pbp1a, and not RodA. Because this point is central to the entire work, it must be tested, either by in vivo controls, or by purification of proteins and testing for binding.

A second large problematic part is the single molecule tracking. Here, the authors just report on mobilities without showing any functionality of fusions or actual microscopy.

- Please state how fusions were made. Which plasmids? N- or C-terminal? Original locus or ectopically?

- Fluorescent images of their DacB, PBP1a1 and PBP1c-PAmCherry fusions are missing, please show examples of cells expressing the fusions, and examples of single molecule tracks.

- Functionality testing of the PBP1a1 and PBP1c-PAmCherry fusions are missing. Please

show that fusions can functionally replace wild type copies and w´ show a Western blot reporting on proper expression levels of fusion.

- If Moenomycin acts by blocking the active site of PBP1a2, it should be tested if introducing an inactive PBP1a2 (mutation) or a reduction of the expression-level has similar effects.

- It is shown that the presence of DacB has an effect on the mobility of PBP1a1 and PBP1c, which would suggest an interaction. Investigation/validation of this possible interaction is missing.

In the absence of these controls, it is impossible to evaluate whether any conclusions drawn are valid.

Reviewer #3 (Remarks to the Author):

Comments on 362212

This is an intriguing paper on the collaboration between class A PBP1a2 and the carboxypeptidase/endopeptidase PBP4 (dacB). *M. xanthus* can be induced by glycerol to sporulate by degrading their PG layer. When they germinate, they first repair their PG and then move the synthesis system away from the future cell poles, causing the cells to elongate. *M. xanthus* uses the rodA/PBP2 couple to elongate and the class A PBPs for repair and assistance in rod growth. When vegetative cells are inhibited by moenomycin (class A PBP GlycosylTransferase inhibitor), the cells also become spherical, whereas mecillinam (transpeptidase inhibitor of PBP2) treated cells remain rod shaped. Deletion of all class A PBPs still allowed rod-shaped growth. This indicated that inhibiting class A PBPs but not their absence triggered PG hydrolysis. DacB or PBP4 appeared to be the responsible candidate. By measuring the localization and dynamics of DacB and the three class A PBP1s, the authors revealed that PBP1a2 and PBP4 together are responsible for the sphere formation in the presence of moenomycin. This activity occurs everywhere in the envelope, but specifically in the poles as these proteins tend to accumulate in the poles. The authors also correlate the diffusion rate and immobile fraction of PBP4 to it being active. In the absence of PBP1a2, the mobility of PBP4 is not affected by moenomycin, indicating that 1a2 support the activity of PBP4 when moenomycin is present. The dynamics and localization of PBP1A2 is not affected by the absence of dacB, whereas PBP1C, becomes much more immobile and polarly localized.

Introduction.

Line 33: "During the vegetative growth of most rod-shaped bacteria, PG is assembled by two major enzymatic systems, the Rod system and class A penicillin-binding proteins (aPBPs)." This is only true if you would consider FtsW/I also the rod system, but that is not what you describe.

Line 43: When the rod system is not working, the cells become spherical and can survive that by producing more FtsZ, PBP1B and PBP3 for example. The remark that the cells remain rod shaped when the rod system is inhibited is misleading. (Unless you do not generalize but immediately mention that you mean in *M. xanthus*).

Line 88: Mecillinam is not inhibiting the Rod system completely, RodA might still be able to produce non-crosslinked glycan strands, which can then be incorporated into the PG layer by class A PBPs?

Because moenomycin is much longer around than the notion that RodA is a GTPase, maybe you should mention specifically that it does not inhibit RodA (Emami, K. et al. RodA as the missing glycosyltransferase in *Bacillus subtilis* and antibiotic discovery for the peptidoglycan polymerase pathway. *Nature Microbiology* 2, 16253–9 (2017)). This ref is already in your ref list, so it does not take extra space.

Figure 2. Why only 20 cells, whereas you can show 4 cells in one view (fig. 2D). Should you not have at least a few hundred cells?

Figure 3, please explain the abbreviation UT.

Since no information is given, I assume that all experiments are based on one single experiment. That does not need to be a problem since the differences between the strains are quite big and consistent. However, it is also not clear whether the 3500 datapoint on the protein dynamics are all derived from 1, 20 of 100 cells. It would be useful to add this information.

I was surprised that the proteins fusion dacB-PAmCherry seemed to be functional as I know that E. coli PBP4 is not functional as a fusion. To understand this, I predicted the structure of M. xanthus dacB and made an overlay with that of E. coli dacB. M. xanthus dacB has an extra domain that seem to block substrate access and it has a completely free C-terminus.

This explains its functionality. Could it also explain how it can be activated by protein interactions? The fact that overproduction of PBP4 is sufficient to make the cells supersensitive to moenomycin, does suggest that it has some activity that does not need protein interactions. Based on figure 3C the cells stay also rod-shaped when deleting single class A PBPs and treating the cells with moenomycin. This is not very well described in the text. In the text one could think that moenomycin was not added. What happens to the localization of PBP4 when moenomycin is added to a triple class A deletion strain? Is it possible that moenomycin directly activates PBP4?

In the included overlay, green is M. xanthus and cyan is E. coli.

The poles are inert with respect to PG synthesis. Your data suggest that they are continuously renewed by the class A PBPs plus dacB. Is it known whether the puling motility requires opening and closing of the PG layer all the time?

What happens with the mobility of the classA PBPs in the presence of moenomycin? Class A PBP1a2 localization is not at all affected by the absence of PBP4, whereas 1C is changing its behavior dramatically. Does this suggest that 1a2 is inactive in the absence of dacB?

In the materials and methods, the construction and the sequence of the chromosomal fluorescent protein fusions is absent.

Although the observations are interesting, I miss experiments that explain what the role is of these proteins in the cell poles. If dacB is activated by 1a2, why does its overexpression then have such a huge effect on moenomycin resistance?

Reviewer #3 Attachment on the following page.

Responses to reviewers' comments

Underlined texts are directly copied and pasted from the revised manuscript. Their line numbers in the manuscript are also provided.

Reviewer #1 (Remarks to the Author):

In this manuscript, the authors explore mechanisms involved in moenomycin resistance in *Myxococcus xanthus*. From other bacteria, it is known that moenomycin inhibit the glycosyltransferase activity of aPBPs involved in peptidoglycan (PG) synthesis thereby causing cell lysis. Here, the authors find that moenomycin sensitivity depends on DacB, a D-Ala-D-Ala carboxypeptidase. Moreover, it is reported that inhibition of PBP1a2 by moenomycin may activate DacB thereby causing excessive PG hydrolysis and cell lysis. By contrast, lack of PBP1a2 is reported to not activate DacB.

This is a potentially highly interesting manuscript, which may help to elucidate how moenomycin causes cell lysis. However, as outlined in details below, the manuscript suffers from the authors not including the original data in their figures. Also, no statistical test to evaluate for significant differences are included. Also, simple control experiments and follow up experiments are missing.

Major comments:

1. line 58-61: Description of aPBPs in *M. xanthus*: From the description of the aPBPs it is not clear whether they are essential. In other words, please clarify whether the rod-shaped cells turned spherical are viable. Also include details on how the aPBPs were inhibited.

We agree with the reviewer. Following the reviewer's suggestion, we changed the description to, "In response to chemical signals, such as glycerol, vegetative cells of *M. xanthus* thoroughly degrade their PG and transform into spherical spores within a few hours²¹⁻²³.

When spores germinate, emerging cells rebuild rod-shaped PG without preexisting templates²³. Such morphological transitions make *M. xanthus* an ideal organism to study PG degradation and assembly²⁴. The Rod system is essential for the *de novo* establishment of rod shape during *M. xanthus* spore germination^{23,25}. In contrast, spherical spores can still germinate into rods in the presence of cefsulodin and cefmetazole, the inhibitors of aPBPs. However, these newly established rods retrogress to spheres before cell division²³. This observation suggests that aPBPs are important for the maintenance of rod-like morphology albeit the mechanism remains unknown." (L57-67).

2. Line 78: Do all the moenomycin treated cells eventually become spherical and lyse? Please include these data to clarify.

Inspired by the reviewer's comment, we performed new experiments and found that moenomycin does not cause cell lysis. Strikingly, the moenomycin-induced spheres can divide into daughter cells of the same size. The reviewer's suggestion has started a new project in our lab: to investigate how *M. xanthus* cells divide as spheres.

Additional data are presented in Fig. 1A, 1B. And these results are described as, "Although prolonged moenomycin treatment turned 100% cells (n > 1000) into spheres, we did not observe mass cell lysis. Instead, cell concentration continued to increase during the treatment (Fig. 1A, 1B). Different from L-forms that usually generate new cells through irregular budding²⁹, the moenomycin-induced spheres produced daughter cells of the same

size (Fig. 1A), indicating that these cells proliferate through binary fission. Currently we are investigating how *M. xanthus* cells divide as spheres (L92-98).

2. Line 84: How are spores formed in untreated cells? Please clarify.

We changed the description to, “Compared to the glycerol-induced spores...” (L89).

3. Line 94 and figure 1B, figure 2C: What is meant by “delay in growth”? Please show growth curves for all the mutants, so that the growth defect can be clearly seen. Please also include tests for significant difference and number of replicates analyzed.

Following the reviewer’s suggestion, we presented the growth curves for these strains and provided significance test. The word “delay” was replaced by “slow-down of growth in early exponential phase” (L110).

4. Line 95 and figure 1C: Please include data on cell length and cell width for all the relevant mutants to document the statement that the mutants maintain rod-shape. Please also include tests for significant differences and number of replicates analyzed.

We appreciate the reviewer’s suggestion. We measured the dimensions of wt and mutant cells and found, “all seven mutants maintained rod shape. Nevertheless, compared to wild-type, mutant cells were moderately, but significantly shorter and wider (Fig. 1D). Thus, while being dispensable for both growth and rod-like morphology, all aPBPs regulate cell dimensions” (L110-113).

5. line 110-116: Why are 2µg/ml of moenomycin used in this experiment and 4µg/ml in the experiments in figure 1. This makes a comparison difficult. Also for these experiments: Please show growth curves in figure 2C. Moreover, to strengthen the idea that lack of DacB gives resistance to moenomycin, please include data to show the MIC of the WT, the dacB mutant and the pbp1a2 mutant (see below), and the strain overexpressing dacB. Also, please include quantitative data on cell morphology of these different strains. These data are essential to evaluate the connection between pbp1a2, dacB and moenomycin.

We thank the reviewer for pointing out this mistake, which has been corrected in the revised manuscript. 8 µg/ml was used for to investigate the effect of moenomycin on cell morphology. For all the fluorescence imaging experiments, moenomycin was used at 4 µg/ml. And the concentrations of antibiotics were stated in the Materials and Methods.

As pointed out in the answer to question #2, moenomycin does not cause mass cell lysis. For this reason, plotting growth curves may not be the best way to quantify the sensitivity to moenomycin. To solve this problem, we treated cells for 2 h using 8 µg/ml moenomycin and quantified the percentage of cells that remained rod-shaped (Fig. 2C).

Cell morphologies of these strains are shown in Fig. 2C.

6. line 117-123 and figure 2D: Please include before and after images of the moenomycin treated cells in 2D. The standard deviations in the diagrams in 2D, right are large. Are any of the observed differences significant? Please also include tests for significant difference and number of replicates analyzed.

Fig. 2A presents the images before moenomycin treatment. To simplify the data presentation, we only showed the TADA fluorescence intensities at 0°, 90°, 180°, and 270°. The description regarding statistical analysis was added to the legend of Fig. 1., “Here and in all subsequent figures, data were pooled from three independent experiments and *p* values were calculated

using the Student paired t test with a two-tailed distribution”. The numbers of replicates were included in the original figure and figure legend.

7. line 124-130 and figure 2c: Please include a western to show that DacB-mCherry accumulates as a full-length fusion protein. These data are essential to be able to evaluate all the localization analyzes.

The Western blot of DacB-PAmCherry and DacB^{S75A}-PAmCherry was presented in Fig. S1.

8. line 124-130 and figure 2c and figure 3: If the full-length fusion protein accumulates, then please include data from simple regular low-resolution epifluorescence microscopy to show the overall localization of DacB-mCherry in the different strains analyzed and +/- moenomycin. These data are essential to be able to evaluate whether the single molecule tracking reflects the overall DacB-mCherry distribution/localization.

We composited 100 frames of single-molecule DacB-PAmCherry in each genetic background and antibiotic treatment into one single image, which depicts the overall localization of DacB. Such information is shown in Fig. S2, S4.

9. All the single molecule tracking experiments: Please include a vector control, i.e. mCherry without DacB, to document that the particles tracked are the fusion protein and not random background. Also, to be able to evaluate the tracking data, please include trajectories of e.g. 10 particles per subset. Also, try to superimpose all the trajectories for a given experiment on a schematic cell. These data are essential to be able to evaluate the various statements. Many data are reported on mobile/immobile fractions and diffusion coefficients. For all these numbers, there are no tests included for significant differences and number of experiments. Please also include tests for significant differences and number of replicates analyzed. Also, please make clear which concentration of moenomycin is used in the different experiments. In a previous report (Fu et al., 2018, PNAS), we reported that due to their extremely fast diffusion, free PAmCherry particles could not be imaged at 10 HZ, the frequency we used for tracking DacB-PAmCherry. Free PAmCherry became traceable when the imaging frequency was increased to 67 Hz. In the revised manuscript, we mentioned this control, “As free PAmCherry molecules appear as blurry objects that cannot be followed at 10-Hz¹⁶, the noise from any potential degradation of DacB-PAmCherry was automatically excluded from our analysis” (L181-183).

Sample individual trajectories are showed in Fig. S2, S4.

Significance test, number of replicates, and the concentration of antibiotics were added into the legends of all the related figures.

10. line 171-174: It is assumed that hydrolytic activity of DacB-mCherry is reflected in the mobility of single particles. No evidence is included to directly support this conclusion. Because the interpretation of many of the presented data rely on this assumption, it is essential to include data to support this assumption. This could for instance done by including the localization of a DacB active site mutant.

We appreciate the reviewer for this important and constructive suggestion. In the first submission, we presented that DacB mobility decreases in moenomycin treated and glycerol-induced sporulating cells, where DacB hydrolase activity is upregulated. In the revised manuscript, we expressed DacB^{S75A}-PAmCherry as the sole source of DacB under the native control of the *dacB* gene. “We expected that the mutation on its active site would prevent

DacB^{S75A}-PAmCherry from binding to PG and thus increase its mobility. Surprisingly, DacB^{S75A}-PAmCherry particles displayed reduced mobility, where 21.2% (n = 2758, compared to 13.1% for DacB-PAmCherry) of particles were immobile and the diffusion coefficient of the mobile population was $2.1 \times 10^{-2} \pm 5.8 \times 10^{-3} \mu\text{m}^2/\text{s}$ (n = 2173, compared to $2.7 \times 10^{-2} \pm 4.1 \times 10^{-3} \mu\text{m}^2/\text{s}$ for DacB-PAmCherry). These data strongly suggested that despite to the loss of peptidase activity, DacB^{S75A} could bind PG with enhanced affinity. To test this hypothesis, we expressed the periplasmic domains (amino acids 21 – 494) of DacB and DacB^{S75A} in *E. coli* and purified the recombinant proteins. After a 1-h incubation with purified *M. xanthus* PG and a brief centrifugation, both proteins precipitated with PG. Compared to the wild-type protein, a significantly bigger population of DacB^{S75A} enriched in the pellet, indicating increased binding to PG (Fig. 3D). The mechanism for this enhanced binding remains to be elucidated. Nevertheless, as multiple residues contribute to PG-binding in DacB³², other residues could overcompensate the substitution of the active serine. Taken together, increased immobile population and decreased diffusion coefficients of DacB particles strongly correlates with their enhanced binding to PG” (L203-219). We believe that the reduced mobility of DacB^{S75A}-PAmCherry has provided yet additional evidence that DacB mobility strongly correlates with its binding to PG.

11. line 180-184: The prediction from the data presented is that a pbp1a2 mutant should be resistant to moenomycin. Is that the case? Also, an active site mutant of PBP1a2 should be resistant to moenomycin and have no effect on DacB. Is that the case? These additional data are important to evaluate the various inferences made by the authors.

We appreciate the reviewer for this important and constructive suggestion. We presented the data of all aPBP mutants and PBP1a2^{E97A} in the revised manuscript. **First**, the E to A mutation on PBP1a2 produced same phenotypes on cell morphology (cells became shorter and wider) as the $\Delta pbp1a2$ strain. However, the E to A mutation did not affect moenomycin resistance, suggesting that PBP1a2^{E97A} still binds moenomycin. **Second**, we found that PBP1a2^{E97A} does not affect the dynamics of DacB, but moenomycin still modulates DacB through PBP1a2^{E97A}. “Like the $\Delta pbp1a2$ cells, the cells expressing PBP1a2^{E97A} enriched DacB molecules at cell poles (Fig. 4A). However, inactivating PBP1a2 through mutagenesis did not reduce DacB mobility (Fig. 4B, 4C), indicating that the binding between moenomycin and PBP1a2 is required to promote PG-binding by DacB (see discussion). Consistently, moenomycin reduced DacB mobility through PBP1a2^{E97A} (Fig. 4B, 4C)” (L254-258). We believe that this experiment strongly supports our conclusion that moenomycin regulates DacB through PBP1a2.

12. line 192- 201 and figure 4: Include westerns to document the accumulation of the various fusion proteins. These data are essential to be able to evaluate all the localization analyzes. If the full-length fusion proteins accumulate, then please include data from simple regular low-resolution epifluorescence microscopy to show the overall localization of the PBPa-mCherry proteins in the different strains analyzed and +/- moenomycin. These data are essential to be able to evaluate whether the single molecule tracking reflects the overall distribution/localization of these proteins.

We appreciate the review for this suggestion. Western blot, overall localization, as well as sample trajectories of all three aPBPs and PBP1a2^{E97A}-PAmCherry are provided in Fig. S4.

Minor comments:

1. line 36: what is meant by “for the majority of PG growth”? Maybe replace with “synthesis during cell growth”. Also, replace “which” with “and”.

Changed following the reviewer’s recommendation.

2. line 39: What does “function outside of…” mean? Function independently?

Changed to, “and their functions do not depend on the Rod system” (L41).

3. line 56: Replace gram with Gram (it is a name).

Changed.

Reviewer #2 (Remarks to the Author):

The coordination between penicillin-binding protein 1 1a (PBP1a) and the hydrolytic peptidase DacB determines the integrity of bacterial cell poles. Zhang et al. have studied the connection between the activity of a class A Penicillin-binding protein, Pbp1a, and a cell wall hydrolase, DacB, in *Myxococcus xanthus*. They propose a model in which PBP1a influences the spatial distribution of DacB within the cell wall, such that inhibition of Pbp1a leads to stronger DacB activity at cell poles, which is detrimental for cells and leads to cell shape changes and ultimately lysis. A regulation of hydrolytic activity via a class A PBP would be a novel mechanism.

However, the study has some serious flaws. Most importantly, it is claimed that the antibiotic “moenomycin” specifically inactivated the transglycosylase of Pbp1a, but not of RodA. This is based on a single publication where a crystal structure of Pbp2 from *E. coli* bound to moenomycin was reported (Lovering et al., *Science* 2007). Because Class A PBPs show quite some sequence divergence, it is not clear if specificity for PBP2 from a different bacterium can be taken face value for specificity for Pbp1a. Additionally, moenomycin is a mimic for a short well wall filament with some modifications, so there is no real reason why it should not bind to RodA. Given that the deletion of the gene encoding for Pbp1a does not result in a change of rod shape, but the depletion of RodA does, and so does addition of moenomycin, it is doubtful whether the chemical really only inhibits Pbp1a, and not RodA. Because this point is central to the entire work, it must be tested, either by in vivo controls, or by purification of proteins and testing for binding.

We appreciate the reviewer for bringing up this question. **First**, as Review #3 kindly pointed out, a previous report (Emami, K. et al. RodA as the missing glycosyltransferase in *Bacillus subtilis* and antibiotic discovery for the peptidoglycan polymerase pathway. *Nature Microbiology* 2, 16253–9 (2017) has clarified that Moenomycin does not inhibit RodA. Accordingly, we changed our description to, “Whereas moenomycin is a phosphoglycolipid that specifically inhibits the GTase activity of aPBPs but does not affect RodA, the GTase of the Rod system⁹, mecillinam is a β -lactam that specifically inhibits the Rod system by blocking the Tase activity of PBP2” (L81-84). **Second**, as mecillinam that inhibits PBP2 and moenomycin show distinct effects on cell morphology (Fig. 1), it is not likely that moenomycin inhibits PBP2 or mecillinam inhibits aPBPs. **Third**, the result that DacB stops responding to moenomycin in the absence of PBP1a2 strongly supports our conclusion that moenomycin regulates DacB through PBP1a2.

A second large problematic part is the single molecule tracking. Here, the authors just report

on mobilities without showing any functionality of fusions or actual microscopy.

- Please state how fusions were made. Which plasmids? N- or C-terminal? Original locus or ectopically?

We agree with the reviewer. In the revised manuscript, **(i)** we provided Western blots to show that all labeled proteins accumulated as full-length (Fig. S1, S6). **(ii)** We also showed that DacB-PAmCherry is fully functional (Fig. 2). **(iii)** The actual images of labeled proteins, including their overall localization and sample particle trajectories, are shown in Fig. S2, S4, S6. **(iv)** All labeled proteins, except the overexpression of DacB-PAmCherry were expressed using their original gene loci and promoters, which is clarified in the text. **(v)** All proteins were labeled on their N-termini. Detailed descriptions were included in the materials and methods section.

- Fluorescent images of their DacB, PBP1a1 and PBP1c-PAmCherry fusions are missing, please show examples of cells expressing the fusions, and examples of single molecule tracks.

Fluorescence images and sample trajectories are shown in Fig. S2, S4, S6.

- Functionality testing of the PBP1a1 and PBP1c-PAmCherry fusions are missing. Please show that fusions can functionally replace wild type copies and w'show a Western blot reporting on proper expression levels of fusion.

We appreciate the reviewer's suggestion. We provided the evidence that "All three fusions accumulated as stable proteins in *M. xanthus* and unlike the aPBP deletion mutants, cells expressing these fusion proteins maintained wild-type cell morphology and sensitivity to moenomycin (Fig. S6A, S6B), indicating that PAmCherry-labeled aPBPs are functional" (L262-265).

- If Moenomycin acts by blocking the active site of PBP1a2, it should be tested if introducing an inactive PBP1a2 (mutation) or a reduction of the expression-level has similar effects. We appreciate the reviewer for this important and constructive suggestion. We presented the data of all aPBP mutants and PBP1a2^{E97A} in the revised manuscript. **First**, the E to A mutation on PBP1a2 produced same phenotypes on cell morphology (cells became shorter and wider) as the $\Delta pbb1a2$ strain. However, this mutation did not affect moenomycin resistance, indicating that PBP1a2^{E97A} still binds moenomycin. **Second**, we found that PBP1a2^{E97A} does not affect the dynamics of DacB, but moenomycin still modulates DacB through PBP1a2^{E97A}. "Like the $\Delta pbb1a2$ cells, the cells expressing PBP1a2^{E97A} enriched DacB molecules at cell poles (Fig. 4A). However, inactivating PBP1a2 through mutagenesis did not reduce DacB mobility (Fig. 4B, 4C), indicating that the binding between moenomycin and PBP1a2 is required to promote PG-binding by DacB (see discussion). Consistently, moenomycin reduced DacB mobility through PBP1a2^{E97A} (Fig. 4B, 4C)" (L254-258). We believe that this experiment strongly supports our conclusion that moenomycin regulates DacB through PBP1a2.

- It is shown that the presence of DacB has an effect on the mobility of PBP1a1 and PBP1c, which would suggest an interaction. Investigation/validation of this possible interaction is missing.

In the absence of these controls, it is impossible to evaluate whether any conclusions drawn are valid.

We appreciate the reviewer for this comment. We are investigating the interactions between

multiple PG synthases and hydrolases. However, we believe that the mechanisms of such interactions are out of scope of the current study. **First**, such interactions could be transient, “As inhibited PBP1a2 activates the hydrolytic activities of DacB and cells overexpressing DacB are more sensitive to moenomycin than the wild-type ones, our results support the hypothesis that PBP1a2 and DacB may only interact transiently in *M. xanthus* and that one PBP1a2 may activate multiple DacB molecules” (L300-304). **Second**, such interactions may require PG. “We speculate that moenomycin mimics a growing PG stand and thus locks PBP1a2 into a catalytic active conformation²⁶. In contrast, PBP1a2^{E97A} is in the catalytic inactive conformation. Under physiological conditions, catalytic active PBP1a2 could recruit DacB to the PG assembly sites, where DacB breaks existing peptide crosslinks and thus facilitates the insertion of newly synthesized murein strands. The fact that moenomycin-bound PBP1a2 recruits DacB to PG, rather than forming a diffusive PBP1a2-DacB complex, suggests that no matter whether these two proteins interact directly, their coordination is mediated by PG” (L287-294). **Third**, our data suggest that DacB could regulate different aPBPs in distinct manners. “In attempt to understand how DacB regulates aPBPs, we compared the effects of DacB and moenomycin. DacB could activate PBP1a1, as the deletion of *dacB* and addition of moenomycin showed similar effects on the distribution of PBP1a1 particles (Fig. 5C). In contrast, DacB could be inhibitory to PBP1a2 and PBP1c, because the absence of DacB and presence of moenomycin exerted opposite effects on the single-particle dynamics of these two aPBPs (Fig. 5A, 5B). Taken together, our data suggest that DacB could regulate different aPBPs in distinct manners. Notably, the polar enrichment of PBP1a2 in the presence of moenomycin may recruit DacB to cell poles and thus promotes PG degradation at poles” (L269-277).

Reviewer #3 (Remarks to the Author):

Comments on 362212

This is an intriguing paper on the collaboration between class A PBP1a2 and the carboxypeptidase/endopeptidase PBP4 (*dacB*). *M. xanthus* can be induced by glycerol to sporulate by degrading their PG layer. When they germinate, they first repair their PG and then move the synthesis system away from the future cell poles, causing the cells to elongate. *M. xanthus* uses the *rodA*/PBP2 couple to elongate and the class A PBPs for repair and assistance in rod growth. When vegetative cells are inhibited by moenomycin (class A PBP GlycosylTransferase inhibitor), the cells also become spherical, whereas mecillinam (transpeptidase inhibitor of PBP2) treated cells remain rod shaped. Deletion of all class A PBPs still allowed rod-shaped growth. This indicated that inhibiting class A PBPs but not their absence triggered PG hydrolysis. DacB or PBP4 appeared to be the responsible candidate. By measuring the localization and dynamics of DacB and the three class A PBP1s, the authors revealed that PBP1a2 and PBP4 together are responsible for the sphere formation in the presence of moenomycin. This activity occurs everywhere in the envelope, but specifically in the poles as these proteins tend to accumulate in the poles. The authors also correlate the diffusion rate and immobile fraction of PBP4 to it being active. In the absence of PBP1a2, the mobility of PBP4 is not affected by moenomycin, indicating that 1a2 support the activity of PBP4 when moenomycin is present. The dynamics and localization of PBP1A2 is not affected by the absence of *dacB*, whereas PBP1C, becomes much more immobile and polarly localized.

Introduction.

Line 33: "During the vegetative growth of most rod-shaped bacteria, PG is assembled by two major enzymatic systems, the Rod system and class A penicillin-binding proteins (aPBPs)." This is only true if you would consider FtsW/I also the rod system, but that is not what you describe.

We appreciate the reviewer for this comment. The description was changed to, "During the vegetative growth of most rod-shaped bacteria, two major enzymatic systems carry out the elongation of PG sacculi" (L34-36).

Line 43: When the rod system is not working, the cells become spherical and can survive that by producing more FtsZ, PBP1B and PBP3 for example. The remark that the cells remain rod shaped when the rod system is inhibited is misleading. (Unless you do not generalize but immediately mention that you mean in *M. xanthus*).

We appreciate the reviewer for this comment. The description was changed to, "While aPBPs assemble PG in different environments, repair PG defects, and regulate cell diameter^{3,11,12}, their absence rarely abolishes cell survival or rod-like morphology^{3,13}. However, the inhibitors of aPBPs usually trigger rapid collapse of rod shapes^{14,15}" (L42-45).

Line 88: Mecillinam is not inhibiting the Rod system completely, RodA might still be able to produce non-crosslinked glycan strands, which can then be incorporated into the PG layer by class A PBPs?

We appreciate the reviewer for this comment. We changed the text to, "It is possible that aPBPs could substitute PBP2 in the presence of mecillinam. Nevertheless, our data indicate that the inhibition of aPBPs by moenomycin is sufficient to trigger rapid collapse of rod shape in *M. xanthus*" (L101-104).

Because moenomycin is much longer around than the notion that RodA is a GTPase, maybe you should mention specifically that it does not inhibit RodA (Emami, K. et al. RodA as the missing glycosyltransferase in *Bacillus subtilis* and antibiotic discovery for the peptidoglycan polymerase pathway. *Nature Microbiology* 2, 16253–9 (2017)). This ref is already in your ref list, so it does not take extra space.

We appreciate the reviewer for this comment, which helps us answer a question from reviewer #1. This information was added to the text, "Whereas moenomycin is a phosphoglycolipid that specifically inhibits the GTase activity of aPBPs but does not affect RodA, the GTase of the Rod system⁹, mecillinam is a β -lactam that specifically inhibits the Rod system by blocking the Tase activity of PBP2" (L81-84).

Figure 2. Why only 20 cells, whereas you can show 4 cells in one view (fig. 2D). Should you not have at least a few hundred cells?

We appreciate the reviewer's suggestion. We have increased the sample size to 300.

Figure 3, please explain the abbreviation UT.

"UT" was changed to "untreated".

Since no information is given, I assume that all experiments are based on one single experiment. That does not need to be a problem since the differences between the strains are quite big and consistent. However, it is also not clear whether the 3500 data points on the protein dynamics are all derived from 1, 20 or 100 cells. It would be useful to add this

information.

We clarified in Materials and Methods, “For each sptPALM experiment, single PAmCherry particles were localized in at least 100 individual cells from three biological replicates...” (L393-395).

I was surprised that the proteins fusion dacB-PAmCherry seemed to be functional as I know that *E. coli* PBP4 is not functional as a fusion. To understand this, I predicted the structure of *M. xanthus* dacB and made an overlay with that of *E. coli* dacB. *M. xanthus* dacB has an extra domain that seem to block substrate access and it has a completely free C-terminus. In the included overlay, green is *M. xanthus* and cyan is *E. coli*.

This explains its functionality. Could it also explain how it can be activated by protein interactions? The fact that overproduction of PBP4 is sufficient to make the cells supersensitive to moenomycin, does suggest that it has some activity that does not need protein interactions.

We appreciate the reviewer for this constructive suggestion. We are currently investigating if DacB directly interacts with PBP1a2. However, we believe that the underlying mechanisms for such functional coordination are more complicated than we have expected. **First**, such interactions could be transient, “As inhibited PBP1a2 activates the hydrolytic activities of DacB and cells overexpressing DacB are more sensitive to moenomycin than the wild-type ones, our results support the hypothesis that PBP1a2 and DacB may only interact transiently in *M. xanthus* and that one PBP1a2 may activate multiple DacB molecules” (L300-304).

Second, such interactions may require PG. “We speculate that moenomycin mimics a growing PG stand and thus locks PBP1a2 into a catalytic active conformation²⁶. In contrast, PBP1a2^{E97A} is in the catalytic inactive conformation. Under physiological conditions, catalytic active PBP1a2 could recruit DacB to the PG assembly sites, where DacB breaks existing peptide crosslinks and thus facilitates the insertion of newly synthesized murein strands. The fact that moenomycin-bound PBP1a2 recruits DacB to PG, rather than forming a diffusive PBP1a2-DacB complex, suggests that no matter whether these two proteins interact directly, their coordination is mediated by PG” (L287-294). **Third**, our data suggest that DacB could regulate different aPBPs in distinct manners. “In attempt to understand how DacB regulates aPBPs, we compared the effects of DacB and moenomycin. DacB could activate PBP1a1, as the deletion of *dacB* and addition of moenomycin showed similar effects on the distribution of PBP1a1 particles (Fig. 5C). In contrast, DacB could be inhibitory to PBP1a2 and PBP1c, because the absence of DacB and presence of moenomycin exerted opposite effects on the single-particle dynamics of these two aPBPs (Fig. 5A, 5B). Taken together, our data suggest that DacB could regulate different aPBPs in distinct manners. Notably, the polar enrichment of PBP1a2 in the presence of moenomycin may recruit DacB to cell poles and thus promotes PG degradation at poles” (L269-277).

Based on figure 3C the cells stay also rod-shaped when deleting single class A PBPs and treating the cells with moenomycin. This is not very well described in the text. In the text one could think that moenomycin was not added. What happens to the localization of PBP4 when moenomycin is added to a triple class A deletion strain? Is it possible that moenomycin directly activates PBP4?

We appreciate the reviewer for these comments. In the revised manuscript, we have remade Fig. 3 and Fig. 4. How the absence of aPBPs affects cell morphology and moenomycin resistance was analyzed in Fig. 1D. We do not believe that moenomycin activates DacB directly because the $\Delta dacB$ strain still shows residue sensitivity to moenomycin. We have

provided strong evidence that DacB no longer responds to moenomycin in the absence of PBP1a2 but still does so in the presence of PBP1a2^{E97A} (Fig. 4). These data especially support our conclusion that moenomycin regulates DacB through PBP1a2.

The poles are inert with respect to PG synthesis. Your data suggest that they are continuously renewed by the class A PBPs plus dacB. Is it known whether the pulsing motility requires opening and closing of the PG layer all the time?

The reviewer brought up a very important point. We still believe that PG remains inert during cell elongation. However, certain stresses, such as antibiotics, could break this inertness. We changed our discussion to, "Recent researches on rod-like morphology have emphasized the roles of the cylindrical section where the Rod system plays major roles in the assembly of PG^{2,19,36}. In contrast, the contribution of cell poles, where PG remains metabolically inert after division, is less appreciated³⁷. As moenomycin induces PG hydrolysis at cell poles, we propose that poles may lose inertness under certain stresses. Particularly, our findings provide an explanation for the lytic effects of aPBP inhibitors" (L311-317).

The twitching motility of *M. xanthus* is powered by type IV pili. While the extension and retraction of pili filaments depend on the assembly of extension and retraction ATPases, respectively, the secretion channels remain stable, which do not require opening and closing of the PG layer. It will be interesting to investigate if the initial assembly of the secretion channels requires PG modification.

What happens with the mobility of the classA PBPs in the presence of moenomycin? Class A PBP1a2 localization is not at all affected by the absence of PBP4, whereas 1C is changing its behavior dramatically. Does this suggest that 1a2 is inactive in the absence of dacB?

We appreciate the reviewer for these comments, which motivated us to postulate how DacB regulates aPBPs. "In attempt to understand how DacB regulates aPBPs, we compared the effects of DacB and moenomycin. DacB could activate PBP1a1, as the deletion of *dacB* and addition of moenomycin showed similar effects on the distribution of PBP1a1 particles (Fig. 5C). In contrast, DacB could be inhibitory to PBP1a2 and PBP1c, because the absence of DacB and presence of moenomycin exerted opposite effects on the single-particle dynamics of these two aPBPs (Fig. 5A, 5B). Taken together, our data suggest that DacB could regulate different aPBPs in distinct manners. Notably, the polar enrichment of PBP1a2 in the presence of moenomycin may recruit DacB to cell poles and thus promotes PG degradation at poles" (L269-277).

In the materials and methods, the construction and the sequence of the chromosomal fluorescent protein fusions is absent.

Such information was added to the Materials and Methods. We also provided Table S1 to summarize the strains we used in this report.

Although the observations are interesting, I miss experiments that explain what the role is of these proteins in the cell poles. If dacB is activated by 1a2, why does its overexpression then have such a huge effect on moenomycin resistance?

We appreciate the reviewer for these comments. We have discussed the possible mechanisms in the revised manuscript. "Notably, the polar enrichment of PBP1a2 in the presence of moenomycin may recruit DacB to cell poles and thus promotes PG degradation at poles" (L276-277). "As moenomycin induces PG hydrolysis at cell poles, we propose that poles may lose inertness under certain stresses. Particularly, our findings provide an explanation for the lytic effects of aPBP inhibitors. As the Rod system is generally excluded

from cell poles⁴⁰, when aPBPs are inhibited, poles become void of PG synthases. The lack of PG repair in combination with the activation of hydrolysis, causes rapid PG degradation at poles, which results in sudden loss of rod shape” (L314-318). We hypothesize that PBP1a2 and DacB may only interact transiently in *M. xanthus* and that one PBP1a2 may activate multiple DacB molecules. This hypothesis was mentioned in the Discussion, “As inhibited PBP1a2 activates the hydrolytic activities of DacB and cells overexpressing DacB are more sensitive to moenomycin than the wild-type ones, our results support the hypothesis that PBP1a2 and DacB may only interact transiently in *M. xanthus* and that one PBP1a2 may activate multiple DacB molecules” (L300-304).

Reviewer #1 (Remarks to the Author):

This is a revised version of a manuscript that I have previously seen. The authors have addressed all my comments satisfactorily by including additional experiments, controls & significance tests, and extensive rewriting. This is a highly interesting manuscript, which helps to elucidate how moenomycin causes cell lysis.

Reviewer #2 (Remarks to the Author):

The authors have much improved the manuscript. They have answered all questions regarding functionality and reliability of drug effects. However, it is still hard to follow how single molecule tracking was done.

A) query 9 of reviewer 1: Control for background fluorescence. Authors reply "As free PAmCherry molecules appear as blurry objects that cannot be followed at 10-Hz16, the noise from any potential degradation of DacB-PAmCherry was automatically excluded from our analysis" (L181-183)". This is not correct. A majority of mCherry molecules will be too fast for imaging, but any freely diffusing molecule will also slow down or even stop for some time. Thus, a certain number of mCherry molecules will show up as defined point spread functions even at 100 ms integration time. Thus, there is no automatic exclusion.

However, if there is no free mCherry present in cells (please see comment below), there is no problem with mCherry-based background fluorescence. In this case, it would be helpful to state how many trajectories were obtained in cells lacking any FP fusion ("wild type cells") and how many trajectories were obtained – in the same regime of imaging time – in cells containing FPs.

B) Please show full Western blots including marker sizes, in all supplementary figures containing WBs. It is not clear if there is indeed a band at 28 kDa in the blots as shown.

C) The authors state that single molecule tracking was done as in 3 studies that are cited. However, these studies either do not contain SMT experiments, or they contain single particle tracking experiments, where several molecules appear as a point spread function, instead of single molecules. It is not clear from the present study how authors arrived at defined diffusion constants; they state how they defined immobile and mobile molecules: "The immobile particles remained within a single pixel (160 nm × 160 nm) before photo-bleach and the mobile ones displayed typical diffusive behavior (Fig. S2)." For how long did molecules remain within one pixel, and what is a typical diffusive behavior?

- please show bleaching step analyses to prove that work was done at single molecule level
- please provide information which method was used to determine diffusion constants, and which tests to ensure that static and mobile motion is based on Brownian diffusion and not on stochastic blinking behavior.

Reviewer #2 (Remarks to the Author):

The authors have much improved the manuscript. They have answered all questions regarding functionality and reliability of drug effects. However, it is still hard to follow how single molecule tracking was done.

We appreciate the reviewer's encouraging comments. Please see our point-to-point responses in below.

A) query 9 of reviewer 1: Control for background fluorescence. Authors reply "As free PAmCherry molecules appear as blurry objects that cannot be followed at 10-Hz¹⁶, the noise from any potential degradation of DacB-PAmCherry was automatically excluded from our analysis" (L181-183). This is not correct. A majority of mCherry molecules will be too fast for imaging, but any freely diffusing molecule will also slow down or even stop for some time. Thus, a certain number of mCherry molecules will show up as defined point spread functions even at 100 ms integration time. Thus, there is no automatic exclusion.

We agree with the reviewer. Yes, freely diffusing PAmCherry particles can still be detected if they slow down or stop. However, we only analyzed the particles that remained in focus for 0.4 - 1.2 s. As free PAmCherry particles rarely stayed in focus for over 0.4 s, the noise from such particles is negligible in our experimental setting. We've clarified our experimental setting as, "We imaged single DacB-PAmCherry particles at 10 Hz using single particle tracking photo-activated localization microscopy (sptPALM) under highly inclined and laminated optical sheet (HILO) illumination^{25,34,35}. Using this setting, only a thin section of each cell surface that was close to the coverslip was illuminated. To analyze the data, we only chose the fluorescent particles that remained in focus for 4 - 12 frames (0.4 - 1.2 s). As free PAmCherry particles diffuse extremely fast in the cytoplasm, entering and exiting the focal plane frequently, they usually appear as blurry objects that cannot be followed at 10-Hz close to the membrane²⁵. For this reason, the noise from any potential degradation of DacB-PAmCherry was negligible." (L178-186).

However, if there is no free mCherry present in cells (please see comment below), there is no problem with mCherry-based background fluorescence. In this case, it would be helpful to state how many trajectories were obtained in cells lacking any FP fusion ("wild type cells") and how many trajectories were obtained – in the same regime of imaging time – in cells containing FPs.

We happen to have such data on hand. We've added this information in L186-191, "Using a 405-nm laser (0.3 kW/cm², 0.1 s) to activate PAmCherry and a 561-nm laser for imaging, we detected 2250 DacB-PAmCherry particles from 296 cells (7.6 ± 3.7 per cell). In contrast, we only detected 27 particles from 369 cells without activation and 25 particles from 449 unlabeled cells using the same setting. Thus, the noises from the background, and autoblinking, the spontaneous switch-on of PAmCherry, were also negligible".

B) Please show full Western blots including marker sizes, in all supplementary figures containing WBs. It is not clear of there is indeed to band at 28 kDa in the blots as shown.

Marker sizes have been added to Fig S1, S5, and S6.

C) The authors state that single molecule tracking was done as in 3 studies that are cited. However, these studies either do not contain SMT experiments, or they contain single particle tracking experiments, where several molecules appear as a point spread function, instead of single molecules. It is not clear from the present study how authors arrived at defined diffusion constants; they state how the defined immobile and mobile molecules: “The immobile particles remained within a single pixel (160 nm x160 nm) before photo-bleach and the mobile ones displayed typical diffusive behavior (Fig. S2). “For how long did molecules remain within one pixel, and what is a typical diffusive behavior?”

We appreciate the reviewer for pointing out the problem in the references. Data analysis was performed using the method published in Ref #25 and clarified in the text accordingly.

Regardless mobile or immobile, we only chose the particles that remained in focus for 0.4 – 1.2 s (4 to 12 frames) for further analysis, which is stated in both the Results (L181-183) and Materials and Methods (L411-412) sections.

We have expanded the Materials and Methods section to include the categorization of diffusion and the calculation of diffusion coefficients, “Briefly, cells were identified using differential interference contrast images. Single PAmCherry particles inside cells were fit by a symmetric 2D Gaussian function, whose center was assumed to be the particle’s position²⁵. Particles in consecutive frames were considered to belong to the same trajectory when they were within a user-defined distance of 320 nm (two pixels). Particles that explored areas smaller than 160 nm x160 nm (within one pixel) in 0.4 - 1.2 s were considered as immobile. For the mobile particles, mean square displacements (MSDs) were calculated. Time-averaged MSD (TAMSD) of each mobile particle was calculated according to the standard method and fit to $\log(\text{TAMSD}) = \log(4D) + \alpha \cdot \log(\Delta t)$, where D is the diffusion coefficient and Δt is time lapse^{45,46}. Based on a previous simulation, diffusive particles on *M. xanthus* cell surface feature α values between 0 and 1.5²⁵. In untreated cells, most PAmCherry-labeled particles belonged to this category (93.4% for DacB-PAmCherry, $n = 1995$; 89.7% for DacB^{S75A}-PAmCherry, $n = 2173$; 92.7% for PBP1a1-PAmCherry, $n = 1657$; 94.6% for PBP1a2-PAmCherry, $n = 1728$; and 95.1% for PBP1c-PAmCherry, $n = 1280$). For simplicity, we considered all the mobile particles of these proteins displayed typical diffusion and determined their diffusion coefficient (D) from a linear fit to the first four points of the MSD using a simpler formula $\text{MSD} = 4D\Delta t$ ^{9,25}” (L407-423).

- please show bleaching step analyses to prove that work was done at single molecule level

We agree with the reviewer that it is necessary to clarify that the fluorescent particles we analyzed are not necessarily single molecules. Accordingly, we have changed “single molecules” to “single particles”.

In this report, we used the mobile/immobile population distribution and diffusion coefficients to approximate the binding between proteins and the PG substrate. Assuming the activation of PAmCherry is random (which is supposed to be), the number of molecules in each particle does not affect either the population distribution or diffusion coefficients.

First, the number of molecules in each particle does not affect diffusion coefficients. For example, DacB is supposed to function as dimers. Whether one or two molecules are activated in a dimer, the calculated diffusion coefficient describes the diffusion of the dimer. In our case, the reduced diffusion coefficients approximate stronger binding to PG, regardless of the oligomerization states of the proteins we studied.

Second, the number of molecules in each particle does not affect population distribution. For example, let's say 70% of the DacB particles contain only one activated PAmCherry and 30% contain two, and we found 40 particles at cell poles and 80 at nonpolar regions. If we ignore the number of molecules in each particle, the polar/nonpolar ratio is 40:80, or 1:2. If we consider the different among particles, the polar/nonpolar ratio will be $(40 \times 0.3 \times 2 + 40 \times 0.7 \times 1) / (80 \times 0.3 \times 2 + 80 \times 0.7 \times 1)$, which is 52:104, still 1:2. The same logic also applies to the mobile/immobile population distribution and higher oligomerization states.

Based on the above rationale, we believe that bleaching step experiments are not essential for justifying our conclusions.

- please provide information which method was used to determine diffusion constants, and which tests to ensure that static and mobile motion is based on Brownian diffusion and not on stochastic blinking behavior.

For the mobile particles, mean square displacements (MSDs) were calculated. Time-averaged MSD (TAMSD) of each mobile particle was calculated according to the standard method and fit to $\log(\text{TAMSD}) = \log(4D) + \alpha \cdot \log(\Delta t)$, where D is the diffusion coefficient and Δt is time lapse^{45,46}. Based on a previous simulation, diffusive particles on *M. xanthus* cell surface feature α values between 0 and 1.5²⁵. In untreated cells, most PAmCherry-labeled particles belonged to this category (93.4% for DacB-PAmCherry, $n = 1995$; 89.7% for DacB^{S75A}-PAmCherry, $n = 2173$; 92.7% for PBP1a1-PAmCherry, $n = 1657$; 94.6% for PBP1a2-PAmCherry, $n = 1728$; and 95.1% for PBP1c-PAmCherry, $n = 1280$). For simplicity, we considered all the mobile particles of these proteins displayed typical diffusion and determined their diffusion coefficient (D) from a linear fit to the first four points of the MSD using a simpler formula $\text{MSD} = 4D\Delta t$ ^{9,25}" (L412-423).

The reviewer raised a great point. Autoblinking challenges data interpretation for single-molecule microscopy. However, we find that this phenomenon does not affect our tracking experiments significantly. **First**, we only analyze particles that remain in focus for 0.4 – 1.2 s, which filters out short blinking events. **Second**, we do not allow gaps in trajectories, which further reduces the effect of blinking. **Third**, in a control experiment, we detected very few particles that met our analysis criteria when we imaged DacB-PAmCherry without 405-nm activation. In the revised manuscript, we provided this information, "Using a 405-nm laser (0.3 kW/cm², 0.1 s) to activate PAmCherry and a

561-nm laser for imaging, we detected 2250 DacB-PAmCherry particles from 296 cells (7.6 ± 3.7 per cell). In contrast, we only detected 27 particles from 369 cells without activation and 25 particles from 449 unlabeled cells using the same setting. Thus, the noises from the background, and autoblanking, the spontaneous switch-on of PAmCherry, were also negligible” (L186-191).